# ReasonFlux-PRM: Trajectory-Aware PRMs for Long Chain-of-Thought Reasoning in LLMs

**Jiaru Zou**[1][*], **Ling Yang**[2,4][†], **Jingwen Gu**[3][*], **Jiahao Qiu**[2],
**Ke Shen**[4], **Jingrui He**[1], **Mengdi Wang**[2][†]
[1]UIUC [2]Princeton University [3]Cornell University [4]ByteDance Seed
 Code: ReasonFlux-PRM-Code,  Models: ReasonFlux-PRM-1.5B/7B

## Abstract

Process Reward Models (PRMs) have recently emerged as a powerful framework for supervising intermediate reasoning steps in large language models (LLMs). Previous PRMs are primarily trained on model final output responses and struggle to evaluate intermediate thinking trajectories robustly, especially in the emerging setting of trajectory–response outputs generated by frontier reasoning models like Deepseek-R1. In this work, we introduce **ReasonFlux-PRM**, a novel trajectory-aware PRM explicitly designed to evaluate the trajectory-response type of reasoning traces. ReasonFlux-PRM incorporates both step-level and trajectory-level supervision, enabling fine-grained reward assignment aligned with structured chain-of-thought data. We adapt ReasonFlux-PRM to support reward supervision under both offline and online settings, including (i) selecting high-quality model distillation data for downstream supervised fine-tuning of smaller models, (ii) providing dense process-level rewards for policy optimization during reinforcement learning, and (iii) enabling reward-guided Best-of-N test-time scaling. Empirical results on challenging downstream benchmarks such as AIME, MATH500, and GPQA-Diamond demonstrate that ReasonFlux-PRM-7B selects higher quality data than strong PRMs (e.g., Qwen2.5-Math-PRM-72B) and human-curated baselines. Furthermore, ReasonFlux-PRM-7B yields consistent performance improvements, achieving average gains of 12.1% in supervised fine-tuning, 4.5% in reinforcement learning, and 6.3% in test-time scaling. We also release an efficient ReasonFlux-PRM-1.5B for resource-constrained applications and edge deployment.

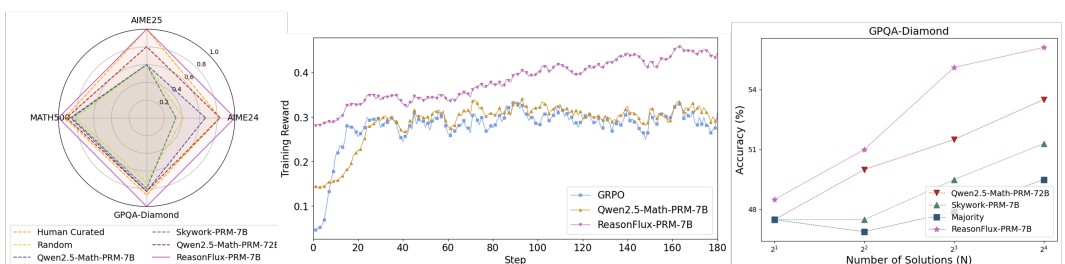

Figure 1: Overview of ReasonFlux-PRM. ReasonFlux-PRM is designed to provide general-purpose reward supervision across multiple application scenarios. **Left:** Offline selection of high-quality distilled trajectory–response data to enhance downstream supervised fine-tuning of smaller models. **Middle:** Online reward modeling integrated into GRPO-based policy optimization. **Right:** Reward-guided Best-of-N test-time scaling to improve inference-time performance.

---

[*]Equal contribution
[†]Corresponding authors

39th Conference on Neural Information Processing Systems (NeurIPS 2025).

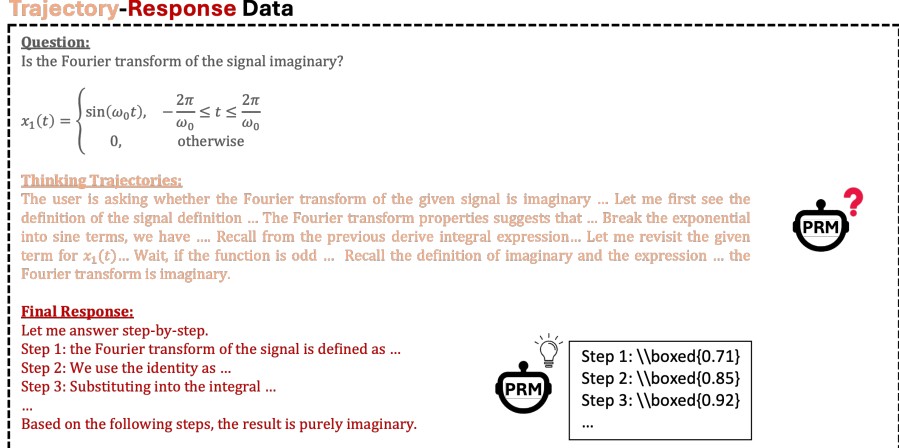

Figure 2: Illustration of the Trajectory-Response Data generated by Deepseek-R1. Existing PRMs can assign appropriate scores to final responses but often struggle to evaluate intermediate reasoning trajectories accurately.

# 1 Introduction

Process Reward Models [49, 79, 24] have recently emerged as a powerful framework for providing process-level supervision in large language models (LLMs) reasoning process, particularly for complex domains such as mathematical problem solving [35, 27, 32, 49, 44]. Given a question and the corresponding model's final response, PRMs verify the reasoning step-by-step and assign fine-grained rewards to each step of the response. Prior studies have leveraged PRMs in both post-training stages [56, 8], including providing dense rewards for online reinforcement learning (RL) [5], and reward-guided inference-time scaling [18, 80].

However, existing PRMs are primarily trained and applied to model-generated final responses, typically presented in an explicit and organized stey-by-step chain-of-thought (CoT) format. Concurrently, with recent advancements in frontier reasoning models such as OpenAI-o1 [16] and Deepseek-R1 [6], these models have increasingly adopted a **trajectory-response** format of output: a lengthy, comprehensive, and less organized intermediate *thinking trajectory*, followed by a concise, step-by-step *final response* conditioned on the prior thinking (as illustrated in Figure 2). Such trajectory–response pairs have been widely distilled and acquired from large reasoning models to support downstream training of smaller models, enabling them to emulate the reasoning capabilities of larger models to first think then produce coherent, extended CoT rationales [40, 72, 68]. The increasing utilization of trajectory–response data raises an important question: **Can PRMs provide supervision not only to the final responses of large reasoning models, but also to their intermediate thinking trajectories?**

Addressing this question first presents a challenge of how to assign informative and correct rewards to the model intermediate thinking trajectories. Unlike final responses, these trajectories are typically treated as silver-standard data [73], automatically generated by large reasoning models without rigorous quality control or standardized verification criteria, making their evaluation inherently noisy and less reliable. To address this, we first revisit several state-of-the-art PRMs and evaluate their performance on trajectory–response pairs. Our analysis reveals that existing PRMs struggle to robustly supervise model thinking trajectories and can degrade downstream training on such data. We further find that this degradation stems primarily from two key issues: an structural and formatting mismatch between intermediate thinking trajectories and final responses, and the lack of trajectory–response data with assigned rewards during PRMs training.

Motivated by these observations, we propose a new trajectory-aware PRM, namely **ReasonFlux-PRM**, which incorporates both step-level and trajectory-level supervision to better align the models' middle thinking trajectories with their final responses. ReasonFlux-PRM is trained on a 10k curated dataset of high-quality trajectory–response pairs covering math and science reasoning. Unlike existing PRMs, ReasonFlux-PRM is explicitly tailored to intermediate thinking processes by providing fine-grained rewards as supervision signals for each step within the thinking trajectory. We further

adapt ReasonFlux-PRM for more general reward modeling scenarios, as illustrated in Figure 1. In *offline* settings, ReasonFlux-PRM assigns scores to filter high-quality trajectory–response pairs, facilitating effective training data curation for downstream supervised fine-tuning of smaller models. In *online* settings, ReasonFlux-PRM is integrated into reward modeling process to provide fine-grained supervision signals during policy optimization, such as GRPO [51]. Moreover, ReasonFlux-PRM facilitates test-time scaling by evaluating multiple generated responses and selecting the most promising one via a reward-guided Best-of-N strategy.

In summary, our main contributions are:

- **In-Depth Trajectory-Response Data Analysis in Long-CoT Reasoning.** We identify, formulate, and analyze the problem of adapting several existing PRMs to supervise both models' intermediate reasoning trajectories and their final responses, motivated by the increasing prevalence of trajectory–response distillation data in downstream post-training and test-time scaling.
- **Trajectory-aware Reward Modeling for Data Selection, RL and Test-Time Scaling.** We introduce ReasonFlux-PRM, a trajectory-aware process reward model that incorporates both step-level and trajectory-level supervision, enabling fine-grained reward assignment for model thinking trajectories. ReasonFlux-PRM can be integrated into both offline and online workflows for more generalized purposes, including offline selection of high-quality training data, online policy optimization in RL training, and test-time scaling.
- **Extensive Downstream Evaluations.** Across extensive evaluations on challenging reasoning benchmarks, ReasonFlux-PRM demonstrates superior data selection quality at smaller model scales, with ReasonFlux-PRM-7B outperforming strong baselines such as Qwen2.5-Math-PRM-72B [79] and datasets curated by human experts. On tasks such as AIME [38, 37], MATH500 [13], and GPQA-Diamond [46], ReasonFlux-PRM-7B achieves notable average accuracy improvement of 12.1% during supervised fine-tuning, 4.5% during reinforcement learning, and 6.3% during inference test-time scaling.

## 2 Preliminaries

**Trajectory-Response Data.** Let $f_{\text{oracle}}(\cdot)$ denote an oracle model, such as Deepseek-R1, capable of producing structured reasoning traces. Given a complex input prompt $x$, the oracle generates a sequence of intermediate thinking steps followed by a final response. We represent each instance of such data as a tuple $(s, a)$, where $s = (s_1, s_2, \ldots, s_T)$ denotes a thinking trajectory consisting of $T$ intermediate steps, and $a = (a_1, a_2, \ldots, a_T)$ denotes the final response, which can also be structured as a chain-of-thought trace with $T$ formatted and organized steps. For large reasoning models, we assume that both $s$ and $a$ consist of $T$ reasoning steps. This structural alignment reflects the modeling assumption that the final output trace $a$ is generated in a step-by-step manner, strictly conditioned on the preceding intermediate reasoning steps $s$. Both the thinking trajectory and final response are generated auto-regressively by the oracle model, i.e.,

$$s_t \sim f_{\text{oracle}}(x, s_{<t}), \quad a_t \sim f_{\text{oracle}}(x, s, a_{<t}), \tag{1}$$

where $s_{<t} = (s_1, \ldots, s_{t-1})$ and $a_{<t} = (a_1, \ldots, a_{t-1})$ denote the reasoning and answer histories up to step $t$, respectively. In the trajectory-response outputs distillation setting, the full supervision target instance $y$ can be constructed as the concatenation of thinking trajectories and the final response, i.e., $y = s \oplus a$.

**Process Reward Modeling.** Given a trajectory-answer pair $(s, a)$, where both $s = (s_1, \ldots, s_T)$ and $a = (a_1, \ldots, a_T)$ are structured as reasoning traces, the goal of a process reward model is to evaluate each intermediate reasoning step $s_t \in s$ with respect to its utility in achieving a correct and coherent final response. We first define a reference reward function $R_{\text{ref}}$ that provides step-level supervision:

$$r_t = R_{\text{ref}}(s_t \mid x, s_{<t}, a), \tag{2}$$

where $R_{\text{ref}}(\cdot)$ scores the $t$-th step conditioned on the input $x$, the prior thinking trajectory steps, and the full final response $a$. The total reward for the trajectory is then computed by aggregating the step-by-step scores:

$$R_{\text{total}} = \mathcal{A}(r_1, r_2, \ldots, r_T), \tag{3}$$

where $\mathcal{A}(\cdot)$ denotes an aggregation function such as MEAN and SUM. The training objective for PRMs is to learn a scoring function $R_\phi(\cdot)$, parameterized by $\phi$, that approximates the reference

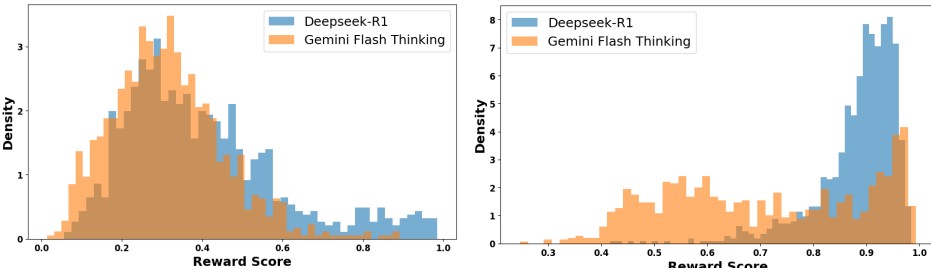

Figure 3: Score distributions rewarded by Qwen2.5-Math-PRM-72B over 1,000 trajectory–response pairs distilled from Deepseek-R1 and the Gemini Flash Thinking API. **Left:** Distribution of scores computed over thinking trajectories. **Right:** Distribution of scores based on final responses.

reward for each step. This is formulated as minimizing the discrepancy between predicted and reference rewards over a training dataset $\mathcal{D} = \{(x^{(i)}, s^{(i)}, a^{(i)}, r_{1:T}^{(i)})\}_{i=1}^{N}$, where $r_t^{(i)}$ denotes the target reward for step $s_t^{(i)}$. Formally, the training objective can be written as:

$$\min_{\phi} \frac{1}{N} \sum_{i=1}^{N} \sum_{t=1}^{T^{(i)}} \mathcal{L}\left(R_{\phi}(s_t^{(i)} \mid x^{(i)}, s_{<t}^{(i)}, a^{(i)}), \ r_t^{(i)}\right). \tag{4}$$

## 3 Existing PRMs Are Not Prepared for Rewarding Thinking Trajectories

To examine whether existing frontier PRMs can be directly applied to reward the trajectory-response data, we first conduct a preliminary study to investigate two key questions:

**RQ1:** Can PRMs distinguish the quality of thinking trajectories distilled from different oracle models?
**RQ2:** What is the effectiveness of using the PRM-selected trajectory-response data on the downstream fine-tuning of smaller models?

For brevity, we defer detailed experimental setups to Appendix A.1. To investigate **RQ1**, we evaluate the Qwen2.5-Math-PRM-72B PRM model on 1,000 sampled problems in s1k [40] with trajectory-response traces generated by Google Flash Thinking API [7] and Deepseek-R1 [6], respectively. For each data trace, we apply the PRM model to compute the step-level rewards (spitted by "\n\n"), and then aggregate these rewards by taking the mean to obtain a final trajectory-level reward. Figure 3 (left) compares the distribution of PRM scores across the two oracle models. The histogram shows a significant overlap in the score distributions, though Deepseek-R1 traces tend to receive higher rewards on average, with a longer tail toward high-reward regions (e.g., scores above 0.6). The results suggest that while Qwen2.5-Math-PRM-72B captures some signal for differentiating between the two sources, its discriminative ability remains limited.

> **Takeaway 1**
>
> Several existing PRMs exhibit limitations in distinguishing reasoning traces distilled from different oracle models and often struggle to clearly separate high- and low-quality model thinking trajectories.

Next, to investigate **RQ2**, we evaluate the performance using the PRM-selected data on the downstream supervised fine-tuning of smaller models. We apply four different PRMs to assign a reward score to each of the 59K raw trajectory-response traces generated by Gemini [7] in s1 [40], using the same mean aggregation over step-level rewards to compute a trajectory-level score. Based on these scores, we rank all traces and select the top 1,000 samples from each PRM as a fine-tuning dataset for the downstream small model. For better comparison, we also adopt the direct set of 1K human-curated examples in s1k [40]. Table 1 presents the accuracy of the fine-tuned Qwen2.5-14B-Instruct on four challenging downstream tasks. We observe that all PRM-selected training sets underperform significantly compared to the human-curated baseline, suggesting that existing PRMs

Table 1: Performance of Qwen2.5-14B-Instruct on four challenging reasoning tasks after fine-tuning on the trajectory-response data selected by four different PRMs. We also compare the SFT performance with using randomly sampled data (1k from 59k) and the s1k human-curated data [40].

| SFT Data Source | AIME24 | AIME25 | MATH500 | GPQA-Diamond |
|---|---|---|---|---|
| Random | 16.7 (↓ 16.6) | 20.0 (↓ 13.3) | 68.4 (↓ 10.4) | 34.8 (↓ 6.6) |
| Math-Shepherd-PRM-7B | 13.3 (↓ 20.0) | 6.7 (↓ 26.6) | 67.8 (↓ 11.0) | 33.3 (↓ 8.1) |
| Skywork-PRM-7B | 13.3 (↓ 20.0) | 13.3 (↓ 20.0) | 71.8 (↓ 7.0) | 37.9 (↓ 3.5) |
| Qwen2.5-Math-PRM-7B | 26.7 (↓ 6.6) | 20.0 (↓ 13.3) | 73.2 (↓ 5.6) | 39.4 (↓ 2.0) |
| Qwen2.5-Math-PRM-72B | 33.3 (↓ 0.0) | 26.7 (↓ 6.6) | 77.0 (↓ 1.8) | 39.4 (↓ 2.0) |
| *on model responses* | 36.7 (↑ 3.4) | 26.7 (↓ 6.6) | 77.8 (↓ 1.0) | 40.9 (↓ 0.5) |
| Human-curated (s1k) | 33.3 | 33.3 | 78.8 | 41.4 |

are not yet sufficiently calibrated to identify high-quality trajectory-response data, and can even degrade downstream model performance by selecting suboptimal or misaligned training samples.

> **Takeaway 2**
>
> Direct reliance on current PRMs for trajectory-response selection can yield misaligned training data, which in turn diminishes the effectiveness of downstream supervised fine-tuning for smaller models.

As most existing PRMs are trained on reasoning traces derived from model final output responses rather than intermediate thinking trajectories [79, 10], we take a closer look at the distinctions between genuine thinking trajectories and post-hoc generated responses. As we detailed in the Appendix A.2, these two types of data exhibit several fundamental differences: *(i) Thinking trajectories often include branching*, where the model revisits earlier steps, explores alternative paths, and revises prior assumptions—behavior rarely observed in the linear and polished structure of final responses. *(ii) Thinking trajectories tend to exhibit weaker global coherence across steps*, as each step is often locally focused and not optimized for narrative continuity.

To further validate that the performance degradation of existing PRMs stems from the aforementioned data mismatch, we conduct an additional experiment in which Qwen2.5-Math-PRM-72B is applied to score each data instance based solely on the model response, rather than the middle thinking trajectories. As shown in Figure 3 (right), the PRM produces a relatively clearer separation in score distributions between the two oracle models. Also as shown in Table 1 (row: on model responses), the performance drop is reduced when training on PRM-selected data based on final responses, suggesting that existing PRMs are better aligned with model-response-level supervision.

> **Takeaway 3**
>
> Thinking trajectories instinctively differ from final responses, and existing PRMs are more accustomed to scoring final outputs than intermediate reasoning steps.

**Motivation on ReasonFlux-PRM.** Our findings above highlight the need for a more general reward model that can effectively evaluate both intermediate model thinking trajectories and final responses. As thinking trajectories become integral to supervised and RL-based fine-tuning, existing PRMs, trained primarily on final responses, struggle to provide reliable supervision. To address this, we propose a new thinking-aware PRM tailored to the trajectory-response data supervision.

## 4 ReasonFlux-PRM

In this section, we introduce ReasonFlux-PRM, a trajectory-aware process reward model, as illustrated in Figure 4. We first present a new reward design tailored for thinking trajectories in Section 4.1, which incorporates both step-level and trajectory-level signals to reflect fine-grained and holistic reasoning quality. We then elaborate how ReasonFlux-PRM is applied in a more general reward supervision setting in Section 4.2, covering both offline data selection and online reward modeling.

Figure 4: Illustration of the overall method design. ReasonFlux-PRM is trained on trajectory–response data pairs with a novel reward design that integrates both step-level and trajectory-level signals. As a general-purpose PRM, ReasonFlux-PRM supports both offline data selection for small models SFT and online reward modeling including policy optimization in RL training and test-time scaling.

## 4.1 How Should We Define Process Rewards and Why?

We first propose a new reward design to train ReasonFlux-PRM from the trajectory–response data. Our formulation integrates both step-level and trajectory-level rewards to better address the discrepancy between intermediate thinking trajectories and final responses, and to align ReasonFlux-PRM with the underlying thinking process through more targeted reward signals during training.

**Step-level reward for thinking trajectories.** As discussed in Section 3, we observe that thinking trajectories are often more complex than final responses, frequently involving branching logic, self-corrections, and redundant reasoning. To better align these two, we incorporate a straightforward *alignment score* $r_t^{\text{align}}$ that measures the semantic similarity between each step in the intermediate thinking trajectories $s_t$ and each step in the final response $a_t$:

$$r_t^{\text{align}} = \text{sim}(\Phi(s_t), \Phi(a_t)), \tag{5}$$

where $\Phi$ is a pretrained encoder and $\text{sim}(\cdot, \cdot)$ denotes cosine similarity. This alignment score uses the final response as a learning signal for earlier thinking trajectories, encouraging those that are topically relevant to the final response and penalizing hallucinated or off-topic content.

Concurrently, to avoid over-penalizing complex yet meaningful thinking trajectory steps that may not be semantically aligned with the final response, we incorporate a complementary *quality score* $r_t^{\text{qual}}$. Inspired by the LLM-as-a-judge paradigm [84, 2, 47], we employ a strong expert model (e.g., GPT-4o) as a judge $J$ to evaluate the logical soundness of each step $s_t$ in context:

$$r_t^{\text{qual}} = J(s_t \mid x, s_{<t}, a). \tag{6}$$

The quality score is designed to capture deeper aspects inside reasoning traces, including step correctness, internal coherence, and progression toward the final response.

In addition to alignment with the final model output and logical step quality, we apply a step-by-step *coherence score* $r_t^{\text{coh}}$ to ensure contextual compatibility between adjacent reasoning steps using a contrastive mutual information formulation. Specifically, we model the coherence between each thinking trajectory step $s_t$ and its predecessor $s_{t-1}$ by contrasting their embedding similarity against $\mathcal{N}$ negative samples drawn from unrelated trajectories:

$$r_t^{\text{coh}} = \log \frac{\exp(\text{sim}(\Phi(s_{t-1}), \Phi(s_t))/\tau)}{\sum_{s' \in \mathcal{N}} \exp(\text{sim}(\Phi(s_{t-1}), \Phi(s'))/\tau)}, \tag{7}$$

where $\tau$ is the temperature parameter. By penalizing incoherent transitions or topic shifts, the coherence score encourages each step to be semantically and logically consistent with its immediate predecessor while remaining distinct from unrelated or disjoint reasoning steps. Finally, to aggregate the alignment, quality, and coherence scores into a unified reward signal, we apply softmax-based

weighting over the three components:

$$r_t^{\text{step}} = \sum_{k \in \{\text{alig, qua, coh}\}} \text{softmax}(r_t^{\text{ali}}, r_t^{\text{qua}}, r_t^{\text{coh}})_k \cdot r_t^k. \tag{8}$$

**Template-Guided Trajectory-level Reward.** While the step-level rewards offer fine-grained supervision on the completeness and coherence of individual reasoning steps, they might not fully assess whether the overall problem-solving strategy encoded in model's thinking trajectory is reliably leads to correct solutions, derived from the final response. We thus introduce a *template-guided trajectory-level reward* to evaluate each trajectory-response data at a higher level of abstraction [69, 68].

Specifically, given an input problem $x$ and the distilled trajectory-response $y = s \oplus a$, we employ a strong expert LLM (e.g., GPT-4o) as a verifier $v$. The verifier processes the complete output $y$ and extracts a reasoning template $\mathcal{T}$, which captures the high-level strategy underlying the original trajectory-response trace. By abstracting the high-level strategy, the template provides a structured guide for subsequent reasoning. The detailed prompt used for template generation is provided in Appendix B. Next, a policy model $\pi_\theta$ is conditioned on the extracted template $\mathcal{T}$ and tasked with solving the input problem $x$ by strictly adhering to the prescribed template $\mathcal{T}$. The model generates $N$ chain-of-thought responses as follows:

$$y^{(1)}, \dots, y^{(N)} \sim \pi_\theta(\cdot \mid x, \mathcal{T}).$$

Then, we define the trajectory-level reward $r^{\text{final}}$ as the average correctness of the generated responses:

$$r^{\text{final}} = \frac{1}{N} \sum_{j=1}^{N} \mathbb{I}(y^{(j)} \text{ is correct}). \tag{9}$$

The template-guided trajectory-level reward evaluates whether the high-level reasoning strategy can be generalized and executed by the policy model independent of the low-level execution in the original trace.

**Joint Training Objective.** To fully leverage both step-level and trajectory-level supervision signals, we integrate the previously defined rewards and propose the following joint training objective:

$$\mathcal{L}_{\text{total}} = \lambda_{\text{step}} \cdot \frac{1}{T} \sum_{t=1}^{T} \mathcal{L}_{\text{step}}\left(R_\phi(s_t \mid x, s_{<t}, a), \ r_t^{\text{step}}\right) + \lambda_{\text{final}} \cdot \mathcal{L}_{\text{final}}\left(R_\phi(x, y), \ r^{\text{final}}\right), \tag{10}$$

where we adopt mean squared error (MSE) as the loss function for both the step and trajectory reward supervision, and $\lambda_{\text{step}}$ and $\lambda_{\text{final}}$ are tunable parameters to balance the relative contributions of fine-grained step supervision and high-level strategic feedback. We train ReasonFlux-PRM with this joint objective as the practical surrogate for the optimization objective in Eq. 4 to align with both token-level and trajectory-level reward signals, thereby enabling the supervision effectiveness on the trajectory-response data.

## 4.2 Offline Data Selection and Online Reward Modeling

We elaborate on the utilities of ReasonFlux-PRM from two perspectives: (i) Offline trajectory-response data selection, where ReasonFlux-PRM is used to identify and select high-quality reasoning traces for downstream supervised fine-tuning and reinforcement learning; and (ii) Online reward modeling, where ReasonFlux-PRM provides token-level and trajectory-level reward signals during RL training, and enables efficient reward estimation for test-time scaling.

**Offline Data Selection.** For offline data selection, ReasonFlux-PRM assigns each trajectory–response pair $(x, y = s \oplus a)$ a step-level reward sequence $\{\hat{r}_t^{\text{step}}\}_{t=1}^{T}$ for each reasoning steps and a trajectory-level reward $\hat{r}^{\text{final}}$. The overall score is computed as:

$$\hat{r} = \frac{1}{T} \sum_{t=1}^{T} \hat{r}_t^{\text{step}} + \alpha \cdot \hat{r}^{\text{final}}, \tag{11}$$

where $\alpha$ balances the contributions of local and global reward signals. The aggregated score $\hat{r}$ is applied to filter samples for later downstream supervised fine-tuning of smaller models.

**Online Reward Modeling.** We first leverage ReasonFlux-PRM to produce a composite reward signal that guides policy optimization through process-level supervision during reinforcement learning. Specifically, during the RL training, we incorporate ReasonFlux-PRM into the Group Relative Policy Optimization (GRPO) [51]. By default, GRPO optimizes for the outcome-level reward $r_{\text{out}}$, which reflects the task accuracy of the policy $\pi_\theta$ on each training sample. To incorporate process-level supervision from ReasonFlux-PRM, we augment this reward with the PRM-based reward $\hat{r}$ in Eq. 11. Given input $x$ and sampled response $y \sim \pi_\theta(\cdot \mid x)$, the new composite reward used for policy training after incorporating ReasonFlux-PRM then becomes:

$$r_{\text{new}} = (1 - \beta) \cdot r_{\text{out}} + \beta \cdot \hat{r}, \tag{12}$$

where $\beta$ controls the relative weight of supervision from $\hat{r}$. With a total of G group size (i.e., number of sampled responses per input), we proceed with group-normalized advantage estimation as:

$$A_{\text{new}} = \frac{r_{\text{new}} - \text{mean}(\{r_{\text{new}}\}_{j=1}^G)}{\text{std}(\{r_{\text{new}}\}_{j=1}^G)}. \tag{13}$$

With the ReasonFlux-PRM derived advantage term $A_{\text{new}}$, we then update the GRPO objective by:

$$\mathcal{J}_{\text{ReasonFlux-PRM-GRPO}}(\theta) = \mathbb{E}_{x_i, \{y_i\}_{i=1}^G \sim \pi_{\theta_{\text{old}}}(\cdot \mid x_i)} \left[ \frac{1}{G} \sum_{i=1}^G \frac{1}{|y_i|} \sum_{t=1}^{|y_i|} \Big( \min \Big\{ \frac{\pi_\theta(y_{i,t} \mid x_i, y_{i,<t})}{\pi_{\theta_{\text{old}}}(y_{i,t} \mid x_i, y_{i,<t})} A_{\text{new}_i}, \right.$$
$$\left. \text{clip}\left( \frac{\pi_\theta(y_{i,t} \mid x_i, y_{i,<t})}{\pi_{\theta_{\text{old}}}(y_{i,t} \mid x_i, y_{i,<t})}, 1 - \epsilon, 1 + \epsilon \right) A_{\text{new}_i} \Big\} - \delta D_{\text{KL}}(\pi_\theta \| \pi_{\text{ref}}) \Big) \right]. \tag{14}$$

Note that ReasonFlux-PRM can be seamlessly integrated into other online RL policy optimization algorithms such as PPo [48] and Reinforce++ [15] by replacing the reward signal with ReasonFlux-PRM 's composite rewards.

**Reward-guided Test-Time Scaling.** During inference, we further apply ReasonFlux-PRM into test-time-scaling strategies such as Best-of-N to identify the most promising output from a set of generated candidates. For each new input question and its corresponding set of sampled model responses, ReasonFlux-PRM assigns a score to each response based on the formulation in Eq. 11, and selects the response with the highest score as the final output.

## 5 Empirical Evaluations

We empirically evaluate ReasonFlux-PRM, focusing on two core applications: (i) Offline data selection, where ReasonFlux-PRM identifies high-quality reasoning traces to improve supervised fine-tuning; and (ii) Online reward modeling, where ReasonFlux-PRM offers reward signals for Best-of-N decoding strategy in test-time scaling and GRPO-based policy optimization.

**Benchmarks.** We evaluate ReasonFlux-PRM on four representative and challenging reasoning benchmarks, including MATH500 [13], a diverse set of 500 mathematical problems of varying difficulty; AIME24 [38], consisting of 30 problems from the 2024 American Invitational Mathematics Examination (AIME); AIME25, which includes 15 problems from the 2025 AIME [37]; and GPQA-Diamond [46], a benchmark of 198 PhD-level science questions to assess advanced scientific reasoning.

**Implementation Details.** We train ReasonFlux-PRM using two off-the-shelf base models, Qwen2.5-1.5B-Instruct and Qwen2.5-7B-Instruct [54], resulting in ReasonFlux-PRM-1.5B and ReasonFlux-PRM-7B, respectively. The training data is primarily sourced from the public trajectory-response reasoning traces such as OpenThoughts-114K [53]. All experiments are conducted on 8 A100 GPUs. Additional experimental setups including ReasonFlux-PRM training details and downstream tasks model configurations are provided in Appendix C.

**Baselines and Models.** For offline data selection, we compare ReasonFlux-PRM with the four frontier PRMs introduced in Section 3, using Qwen2.5-14B-Instruct[67, 54] as the generator model for standard supervised fine-tuning evaluations. For online reward modeling, constrained by computational resources, we primarily use 7B-scale models as policy models for reinforcement learning, including Qwen2.5-7B and Deepseek-R1-Distill-Qwen-7B [6]. For test-time Best-of-N scaling, we adopt Qwen2.5-14B as the generator model to evaluate inference-time performance.

Table 2: Offline Data Selection Comparison. We fine-tune the generator model Qwen2.5-14B-Instruct using data selected by ReasonFlux-PRM-7B and additional baselines. The highest performance of the generators trained on each data source is bold. ReasonFlux-PRM-7B achieves better performance than the strongest human-curated baseline.

| SFT Data Source | AIME24 | AIME25 | MATH500 | GPQA-Diamond |
|---|---|---|---|---|
| Human-curated (s1k) | 33.3 | **33.3** | 78.8 | 41.4 |
| Random | 16.7 (↓ 16.6) | 20.0 (↓ 13.3) | 68.4 (↓ 10.4) | 34.8 (↓ 6.6) |
| Math-Shepherd-PRM-7B | 13.3 (↓ 20.0) | 6.7 (↓ 26.6) | 67.8 (↓ 11.0) | 33.3 (↓ 8.1) |
| Skywork-PRM-7B | 13.3 (↓ 20.0) | 13.3 (↓ 20.0) | 71.8 (↓ 7.0) | 37.9 (↓ 3.5) |
| Qwen2.5-Math-PRM-7B | 26.7 (↓ 6.6) | 20.0 (↓ 13.3) | 73.2 (↓ 5.6) | 39.4 (↓ 2.0) |
| Qwen2.5-Math-PRM-72B | 33.3 (↓ 0.0) | 26.7 (↓ 6.6) | 77.0 (↓ 1.8) | 39.4 (↓ 2.0) |
| *on model responses* | 36.7 (↑ 3.4) | 26.7 (↓ 6.6) | 77.8 (↓ 1.0) | 40.9 (↓ 0.5) |
| **ReasonFlux-PRM-7B** | **40.0** (↑ 6.7) | **33.3** (↑ 0.0) | **84.8** (↑ 6.0) | **47.5** (↑ 6.1) |

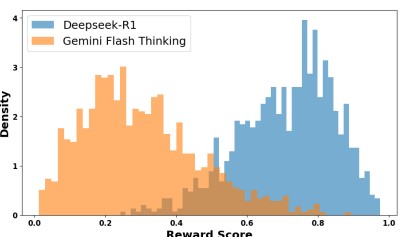

Figure 5: Score distributions rewarded by ReasonFlux-PRM-7B on Deepseek-R1 and Gemini over 1000 trajectory-response data.

Table 3: Performance of PRMs as reward signals in policy optimization. For each of the two policy models, i.e. DeepSeek-R1-Distill-Qwen-7B and Qwen2.5-7B-Instruct, we run GRPO with three different reward signals: entirely rule-based, Qwen2.5-Math-PRM-7B, and ReasonFlux-PRM. The latter two non-rule-based rewards are factored into the overall reward signal according to Eq. 12. We report the mean accuracy averaged over 16 independent runs.

| Policy Model | Reward Signal Source | AIME24 | AIME25 | MATH500 | GPQA-Diamond |
|---|---|---|---|---|---|
| Qwen2.5-7B-Instruct | Rule-based | 12.9 | 11.1 | 73.6 | 32.7 |
| | Qwen2.5-Math-PRM-7B | 12.9 | 13.3 | 74.8 | 32.4 |
| | ReasonFlux-PRM-7B | **16.3** | **17.1** | **77.2** | **34.9** |
| DeepSeek-R1-Distill-Qwen-7B | Rule-based | 50.2 | 38.3 | 89.6 | 47.1 |
| | Qwen2.5-Math-PRM-7B | 51.2 | 40.8 | 92.8 | 49.1 |
| | ReasonFlux-PRM-7B | **54.6** | **44.2** | **94.8** | **51.6** |

## 5.1 Offline Data Selection

Table 2 presents the supervised fine-tuning results of Qwen2.5-14B-Instruct, with training data selected by different strategies, including ReasonFlux-PRM-7B, baseline PRMs, and human-curated examples. Notably, ReasonFlux-PRM-7B outperforms the high-quality human-curated s1k dataset. Specifically, our model achieves a 6.0% gain on MATH500 and a 6.1% improvement on GPQA-Diamond relative to the human-curated baseline. We also plot the score distribution over the 1,000 trajectory-response pairs generated by Deepseek-R1 and Gemini, as shown in Figure 5. The clearly separated score distributions in the figure demonstrate that ReasonFlux-PRM-7B effectively distinguishes between the trajectory-response quality generated by different models, providing a reliable reward signal for high-quality data selection.

## 5.2 Online Reward Modeling

**Reward Signal for RL training.** Figure 6 and Table 3 present the training dynamics and downstream reasoning performance after incorporating different reward signals into policy optimization via GRPO. We evaluate two 7B-scale policy models: Qwen2.5-7B-Instruct and DeepSeek-R1-Distill-Qwen-7B. For each model, we compare three reward signal sources: a fully rule-based heuristic following the original GRPO approach, Qwen2.5-Math-PRM-7B, and ReasonFlux-PRM-7B. Across both policy models and all evaluated tasks, ReasonFlux-PRM-7B consistently delivers superior gains over both the rule-based and prior PRM-based reward signals. On Qwen2.5-7B-Instruct, ReasonFlux-PRM-7B improves performance by 3.4% on AIME24 and 5.8% on AIME25 relative to the rule-based baseline. On the stronger DeepSeek-R1-Distill-Qwen-7B model, ReasonFlux-PRM-7B further advances results, raising MATH500 accuracy from 89.6% to 94.8% and GPQA-Diamond from 47.1% to 51.6%. In addition, when directly comparing ReasonFlux-PRM-7B against Qwen2.5-Math-PRM-7B, we observe consistent improvements. For example, a 3.8% gain on AIME25 with Qwen2.5-7B-Instruct and a 2.5% gain on GPQA-Diamond with DeepSeek-R1-Distill-Qwen-7B. These results demonstrate that the high-quality learned reward signals from ReasonFlux-PRM substantially enhance policy

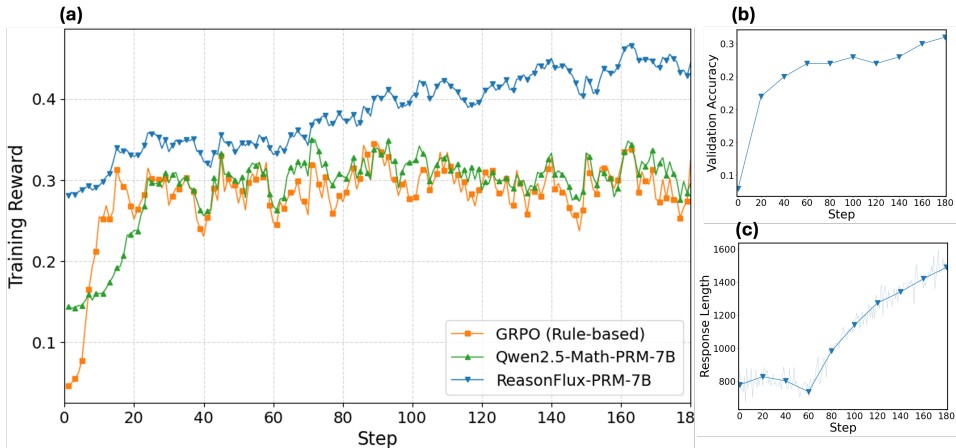

Figure 6: Training dynamics of GRPO policy optimization using ReasonFlux-PRM-7B as reward signals and Qwen2.5-7B-Instrct as the policy model. **(a) Training reward vs. step:** We compare the training reward evolution across original rule-based GRPO, Qwen2.5-Math-PRM-7B, and ReasonFlux-PRM-7B; **(b) Validation accuracy vs. step:** We report the validation accuracy during training with ReasonFlux-PRM-7B; **(c) Response length vs. step:** We report the evolution of generated response lengths over training steps with ReasonFlux-PRM-7B.

optimization, outperforming both heuristic and strong PRM baselines, and ultimately yielding more capable reasoning models through RL training.

**Additional Performance Analyses.** We leave further performance analyses on ReasonFlux-PRM and case studies in Appendix D and Appendix E.

## 6 Related Works

**Process Reward Models.** Process Reward Models (PRMs) [27, 55, 78, 50, 36, 8, 65] provide step-level supervision for model reasoning answers, assigning intermediate rewards to each reasoning step [79, 60, 33, 4, 42]. Existing PRMs, such as Math-Shepherd [59], Skywork-PRM [10], and Qwen2.5-Math-PRM series [79], are trained on either human-annotated rewards [27] or synthesized supervision signals [35] to provide fine-grained step-level rewards for model-generated reasoning solutions across different tasks such as math problem solving [38, 34], science reasoning [46], and programming [12]. More recent work such as Think-PRM [18] introduces a generative PRM to produce long CoT verification. Prior works have integrated PRMs as reward signals during training [56, 49, 23, 8, 85, 82], such as step-by-step verified online RL policy optimization [8, 5, 86, 75, 29] or iterative generator improvement through verifier-guided self-training [14]. Others apply PRMs during inference-time scaling [52, 80, 18, 52, 87, 70, 68, 69] by integrating the models with step-level search and decoding strategies, including beam search [52], reward-guided tree search [62, 43], Best-of-N sampling [28], etc. However, since current PRMs are mostly trained on model-generated final solutions, they struggle to provide effective reward supervision for the internal reasoning trajectories produced by large reasoning models [6, 26] prior to generating final answers. To address this, we design a new trajectory-aware PRM specifically aimed at providing reward supervision for such trajectory–response formatted long CoT data. We leave additional related works in Appendix F.

## 7 Conclusion

We present ReasonFlux-PRM, a trajectory-aware PRM that delivers fine-grained step-level and trajectory-level supervision for trajectory-response long chain-of-thought reasoning traces. Through extensive empirical evaluations, ReasonFlux-PRM consistently improves downstream model performance across multiple challenging benchmarks and application settings. Specifically, ReasonFlux-PRM surpasses strong baselines and human-curated data in offline training data selection, enhances policy optimization during reinforcement learning via dense process-level rewards, and demonstrates superior test-time scaling in Best-of-N inference. Our results highlight the importance of trajectory-aware reward modeling for supervising model intermediate reasoning processes. The discussion of limitations and broader impacts is provided in Appendix G.

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

# Table of Contents

# A   Details on the Preliminary Study in Section 3

## A.1   Preliminary Study Setups

**Process Reward Models.** We evaluate four state-of-the-art process reward models for scoring the quality of the thinking trajectories data: Math-Shepherd-PRM-8B [59], Skywork-PRM-7B [10], Qwen2.5-Math-PRM-7B [79], and Qwen2.5-Math-PRM-72B [79]. The details description for each model is shown below:

- **Math-Shepherd-PRM-8B** [59]: A 7B PRM based on Mistral, trained with data auto-generated from Mistral-7B fine-tuned on MetaMath. It emphasizes verification of step-level reasoning through process-level rewards without human annotations.
- **Skywork-PRM-7B** [10]: A PRM built on Qwen2.5-Math-7B-Instruct and trained on data derived from LLaMA-2 fine-tuned on math tasks. It shows strong generalization for verifying reasoning trajectories across models and supports efficient TTS with low inference overhead.
- **Qwen2.5-Math-PRM-7B** [79]: Trained on Qwen2.5-Math-7B-Instruct using data from the Qwen2.5-Math series, this PRM offers robust step-by-step reward signals and high compatibility with Qwen family models, demonstrating superior supervision ability in TTS tasks among 7B-scale PRMs.
- **Qwen2.5-Math-PRM-72B** [79]: A high-capacity verifier trained on Qwen2.5-Math-72B-Instruct and Qwen-generated data. It achieves state-of-the-art process supervision and excels in guiding both sampling- and search-based TTS strategies across a range of mathematical reasoning tasks.

**Data Sources.** For the data sources, we follow s1k [40] to use its collected datasets consisting of 59K raw model thinking trajectories distilled from the Google Gemini Flash Thinking API [7], along with 1K human-curated samples from the same source and an additional 1K human-curated samples from Deepseek-R1 [6]. These trajectories span a broad range of topics, including math and scientific reasoning. For downstream tasks, we choose 4 challenging benchmarks including: AIME24 [38], AIME25 [37], MATH500 [13], and GPQA-Diamond [46].

**Training Details in RQ2.** As the downstream generator, we adopt Qwen2.5-14B-Instruct as our base model for fine-tuning evaluation. We perform supervised fine-tuning on the Qwen2.5-14B-Instruct model using various 1,000-sample training datasets, each selected either by different PRM-based rankings or curated by human annotators in s1k [40]. We fine-tune the model for 5 epochs using a learning rate of $1e^{-5}$, weight decay of $1e^{-4}$, and a maximum sequence length of 32,768. All experiments are conducted on a server node with 8 A100-80G GPUs.

## A.2   Difference between Model Thinking Trajectories and Final Responses

As we mentioned in Section 3, there are two key difference of the data between model thinking trajectories and final responses:

- **Branching steps across thinking trajectories:** For instance, the thinking trajectories might initially assume an incorrect variable assignment in a math problem, detect the inconsistency, and backtrack to re-derive the solution.
- **Weaker global coherence across steps:** This manifests in speculative or uncertain statements (e.g., "if we assume X, then Y") that may not resolve within the same reasoning path, and in disjointed or redundant logic, such as repeating subgoals or prematurely concluding without fully integrating prior steps. In contrast, final responses are typically globally fluent and logically unified, aiming to deliver a streamlined and conclusive answer.

In Examples 1 and 2 below, we provide detailed data examples that highlight the distinctions between model thinking trajectories and final output responses. These examples showcase how the thinking trajectories often contain exploratory, speculative, or partially resolved reasoning steps, in contrast to the more polished and coherent final outputs. To better illustrate these differences, we highlight all branching steps, backtracking behaviors, and internal deliberations in blue, emphasizing the non-linear and less deterministic nature of the model's intermediate reasoning process.

## Example 1

**Question:** Two capacitors with capacitance values $C_1 = 2000 \pm 10$ pF and $C_2 = 3000 \pm 15$ pF are connected in series. The voltage applied across this combination is $V = 5.00 \pm 0.02$ V.
What is the percentage error in the calculation of the energy stored in this combination of capacitors?

**Thinking Trajectories:**
Let $C_1$ and $C_2$ be the capacitance values of the two capacitors, and $\Delta C_1$, $\Delta C_2$ be their absolute errors.
Given: $C_1 = 2000 \pm 10$ pF $C_2 = 3000 \pm 15$ pF Voltage applied across the combination, $V = 5.00 \pm 0.02$ V When capacitors are connected in series, the equivalent capacitance $C_{\text{eq}}$ is given by:

$$\frac{1}{C_{\text{eq}}} = \frac{1}{C_1} + \frac{1}{C_2} \Rightarrow C_{\text{eq}} = \frac{C_1 C_2}{C_1 + C_2}$$

Let me double check, initially I thought the formula was $C_1 + C_2$, but that's for parallel combination. Since it's in series, we should use the reciprocal formula. Calculate the nominal value:

$$C_{\text{eq}} = \frac{2000 \times 3000}{2000 + 3000} = \frac{6 \times 10^6}{5000} = 1200 \text{ pF}$$

To find the error in $C_{\text{eq}}$, we use:

$$\Delta C_{\text{eq}} = \sqrt{\left(\frac{\partial C_{\text{eq}}}{\partial C_1} \Delta C_1\right)^2 + \left(\frac{\partial C_{\text{eq}}}{\partial C_2} \Delta C_2\right)^2}$$

Partial derivatives:

$$\frac{\partial C_{\text{eq}}}{\partial C_1} = \frac{C_2^2}{(C_1 + C_2)^2}, \quad \frac{\partial C_{\text{eq}}}{\partial C_2} = \frac{C_1^2}{(C_1 + C_2)^2}$$

Evaluate:

$$\frac{9}{25}, \quad \frac{4}{25}$$

$$\Delta C_{\text{eq}} = \sqrt{\left(\frac{9}{25} \times 10\right)^2 + \left(\frac{4}{25} \times 15\right)^2} = \frac{1}{25}\sqrt{90^2 + 60^2}$$

$$= \frac{1}{25}\sqrt{8100 + 3600} = \frac{1}{25}\sqrt{11700} \approx 4.33 \text{ pF}$$

Hmm, but is this the only way? Maybe logarithmic differentiation would give a cleaner expression? Let me try an alternative method:

$$\frac{1}{C_{\text{eq}}} = \frac{1}{C_1} + \frac{1}{C_2} \Rightarrow -\frac{1}{C_{\text{eq}}^2} dC_{\text{eq}} = -\frac{1}{C_1^2} dC_1 - \frac{1}{C_2^2} dC_2$$

So:

$$\frac{dC_{\text{eq}}}{C_{\text{eq}}^2} = \frac{dC_1}{C_1^2} + \frac{dC_2}{C_2^2} \Rightarrow \frac{\Delta C_{\text{eq}}}{C_{\text{eq}}} = C_{\text{eq}}\left(\frac{\Delta C_1}{C_1^2} + \frac{\Delta C_2}{C_2^2}\right)$$

This seems messier, maybe it is generalizable. Now let me compute it:

$$= \frac{3000 \cdot 10}{2000 \cdot 5000} + \frac{2000 \cdot 15}{3000 \cdot 5000} = 0.003 + 0.002 = 0.005 \Rightarrow \Delta C_{\text{eq}} = 0.005 \cdot 1200 = 6 \text{ pF}$$

Now compute energy:

$$E = \frac{1}{2} C_{\text{eq}} V^2$$

Let me double-check: voltage is squared here, so the error contribution from voltage gets doubled in fractional terms. Fractional error:

$$\frac{\Delta E}{E} = \frac{\Delta C_{\text{eq}}}{C_{\text{eq}}} + 2 \cdot \frac{\Delta V}{V} = 0.005 + 2 \cdot 0.004 = 0.013 \Rightarrow \text{Percentage error} = 1.3\%$$

Alternative verification might help... I might try distribute the weight of errors proportionally:

$$\frac{\Delta C_{\text{eq}}}{C_{\text{eq}}} = \frac{\Delta C_1}{C_1} \cdot \frac{C_2}{C_1 + C_2} + \frac{\Delta C_2}{C_2} \cdot \frac{C_1}{C_1 + C_2} = 0.005 \cdot 0.6 + 0.005 \cdot 0.4 = 0.005$$

Let me Final check:

$$\ln E = \ln\left(\frac{1}{2}\right) + \ln C_{\text{eq}} + 2 \ln V \Rightarrow \frac{\Delta E}{E} = \frac{\Delta C_{\text{eq}}}{C_{\text{eq}}} + 2 \cdot \frac{\Delta V}{V}$$

The final answer is:
$$\Rightarrow \boxed{1.3\%}$$

---

**Final Response:**

*Step 1: Calculate the Equivalent Capacitance*

When capacitors are connected in series, the equivalent capacitance $C_{\text{eq}}$ is given by:

$$\frac{1}{C_{\text{eq}}} = \frac{1}{C_1} + \frac{1}{C_2} = \frac{C_1 + C_2}{C_1 C_2} \quad \Rightarrow \quad C_{\text{eq}} = \frac{C_1 C_2}{C_1 + C_2}$$

$$C_{\text{eq}} = \frac{2000 \times 3000}{2000 + 3000} = \frac{6 \times 10^6}{5000} = 1200 \text{ pF}$$

*Step 2: Calculate the Error in $C_{eq}$*

Using the formula for propagation of uncertainty:

$$\Delta C_{\text{eq}} = \left| \frac{\partial C_{\text{eq}}}{\partial C_1} \right| \Delta C_1 + \left| \frac{\partial C_{\text{eq}}}{\partial C_2} \right| \Delta C_2$$

Compute the partial derivatives:

$$\frac{\partial C_{\text{eq}}}{\partial C_1} = \frac{C_2^2}{(C_1 + C_2)^2}, \quad \frac{\partial C_{\text{eq}}}{\partial C_2} = \frac{C_1^2}{(C_1 + C_2)^2}$$

Substitute the given values:

$$\frac{\partial C_{\text{eq}}}{\partial C_1} = \frac{3000^2}{5000^2} = \frac{9}{25}, \quad \frac{\partial C_{\text{eq}}}{\partial C_2} = \frac{2000^2}{5000^2} = \frac{4}{25}$$

Now compute the error:

$$\Delta C_{\text{eq}} = \frac{9}{25} \cdot 10 + \frac{4}{25} \cdot 15 = \frac{90}{25} + \frac{60}{25} = \frac{150}{25} = 6 \text{ pF}$$

*Step 3: Compute the Fractional Error in $C_{eq}$*

$$\frac{\Delta C_{\text{eq}}}{C_{\text{eq}}} = \frac{6}{1200} = 0.005$$

*Step 4: Energy Stored in the Capacitor System*

The energy stored in a capacitor is given by:

$$E = \frac{1}{2} C_{\text{eq}} V^2$$

*Step 5: Propagate Error in Energy Calculation*

The fractional error in energy $E$ is:

$$\frac{\Delta E}{E} = \frac{\Delta C_{\text{eq}}}{C_{\text{eq}}} + 2 \cdot \frac{\Delta V}{V}$$

Given:

$$\frac{\Delta C_{\text{eq}}}{C_{\text{eq}}} = 0.005, \quad \frac{\Delta V}{V} = \frac{0.02}{5.00} = 0.004$$

Substitute into the error formula:

$$\frac{\Delta E}{E} = 0.005 + 2 \cdot 0.004 = 0.005 + 0.008 = 0.013$$

*Step 6: Convert to Percentage Error*

$$\text{Percentage error} = \frac{\Delta E}{E} \times 100\% = 0.013 \times 100\% = \boxed{1.3\%}$$

*Final Answer:* The percentage error in the energy stored is $\boxed{1.3\%}$.

## Example 2

**Question:** One base of a trapezoid is 100 units longer than the other base. The segment that joins the midpoints of the legs divides the trapezoid into two regions whose areas are in the ratio $2 : 3$. Let $x$ be the length of the segment, parallel to the bases, that divides the trapezoid into two regions of equal area. Find the greatest integer that does not exceed $\dfrac{x^2}{100}$.

**Thinking Trajectories:**
The problem asks for the value of $\lfloor x^2/100 \rfloor$, where $x$ is the length of a segment parallel to the bases that divides the trapezoid into two regions of equal area. We are given information about the lengths of the bases and the ratio of the areas created by the segment joining the midpoints of the legs.
Let the lengths of the two bases be $b_1$ and $b_2$, with $b_1 > b_2$. We are told that:

$$b_1 = b_2 + 100$$

Let the height of the trapezoid be $h$. The midline of the trapezoid has length:

$$m = \frac{b_1 + b_2}{2}$$

It divides the trapezoid into two smaller trapezoids, each of height $h/2$. The area of the upper trapezoid is:

$$A_1 = \frac{1}{2}(b_2 + m) \cdot \frac{h}{2} = \frac{h}{8}(b_1 + 3b_2)$$

The area of the lower trapezoid is:

$$A_2 = \frac{1}{2}(b_1 + m) \cdot \frac{h}{2} = \frac{h}{8}(3b_1 + b_2)$$

We are told that the ratio of areas is $2 : 3$ (upper to lower), so:

$$\frac{A_1}{A_2} = \frac{b_1 + 3b_2}{3b_1 + b_2} = \frac{2}{3} \Rightarrow 3(b_1 + 3b_2) = 2(3b_1 + b_2) \Rightarrow 3b_1 + 9b_2 = 6b_1 + 2b_2 \Rightarrow 7b_2 = 3b_1$$

Substitute $b_1 = b_2 + 100$:

$$7b_2 = 3(b_2 + 100) \Rightarrow 7b_2 = 3b_2 + 300 \Rightarrow 4b_2 = 300 \Rightarrow b_2 = 75 \Rightarrow b_1 = 175$$

Let me also test the other case just in case it's a typo, what if the area ratio was actually $3 : 2$?

$$\frac{b_1 + 3b_2}{3b_1 + b_2} = \frac{3}{2} \Rightarrow 2(b_1 + 3b_2) = 3(3b_1 + b_2) \Rightarrow 2b_1 + 6b_2 = 9b_1 + 3b_2 \Rightarrow 3b_2 = 7b_1$$

$$\Rightarrow 3b_2 = 7(b_2 + 100) \Rightarrow 3b_2 = 7b_2 + 700 \Rightarrow -4b_2 = 700 \Rightarrow b_2 = -175$$

That's invalid as the length can't be negative. So the correct ratio is $2 : 3$ and $b_2 = 75$, $b_1 = 175$.
Now consider a segment of length $x$, parallel to the bases, that divides the trapezoid into two equal-area regions. Let the height from $b_2$ up to this segment be $h'$. The total area is:

$$A_{\text{total}} = \frac{1}{2}(b_1 + b_2)h = \frac{1}{2}(175 + 75)h = 125h$$

We want:

$$\frac{1}{2}(b_2 + x)h' = \frac{125h}{2} \Rightarrow (b_2 + x)h' = 125h$$

We also know from similar triangles:

$$\frac{x - b_2}{h'} = \frac{b_1 - b_2}{h} \Rightarrow \frac{h'}{h} = \frac{x - b_2}{b_1 - b_2}$$

Substitute into the area equation:

$$(b_2 + x) \cdot h \cdot \frac{x - b_2}{b_1 - b_2} = 125h \Rightarrow (b_2 + x)(x - b_2) = 125(b_1 - b_2)$$

$$x^2 - b_2^2 = 125(b_1 - b_2) \Rightarrow x^2 = 125(b_1 - b_2) + b_2^2$$

Substitute $b_1 = 175$, $b_2 = 75$:

$$x^2 = 125(100) + 75^2 = 12500 + 5625 = 18125 \Rightarrow \left\lfloor \frac{x^2}{100} \right\rfloor = \left\lfloor \frac{18125}{100} \right\rfloor = \boxed{181}$$

I also saw a different derivation that leads directly to this result. Let me verify it:

$$2x^2 = b_1^2 + b_2^2 \Rightarrow x^2 = \frac{b_1^2 + b_2^2}{2} \Rightarrow x^2 = \frac{175^2 + 75^2}{2} = \frac{30625 + 5625}{2} = 18125 \Rightarrow \left\lfloor \frac{x^2}{100} \right\rfloor = \boxed{181}$$

---

**Final Response:**

*Step 1: Set up variables and use the area ratio condition*

Let the lengths of the two bases of the trapezoid be $b_1$ and $b_2$, with $b_1 > b_2$. We are given that $b_1 = b_2 + 100$.

The segment that joins the midpoints of the legs (the midline) divides the trapezoid into two smaller trapezoids with heights $h/2$. The lengths of the parallel sides of the upper trapezoid are $b_2$ and $m = \frac{b_1 + b_2}{2}$, and its area is

$$A_1 = \frac{h}{8}(b_1 + 3b_2).$$

The lengths of the parallel sides of the lower trapezoid are $m$ and $b_1$, and its area is

$$A_2 = \frac{h}{8}(3b_1 + b_2).$$

The ratio of the areas is $A_1 : A_2 = (b_1 + 3b_2) : (3b_1 + b_2)$. Given that this ratio is $2 : 3$ or $3 : 2$. If $\frac{b_1 + 3b_2}{3b_1 + b_2} = \frac{2}{3}$, then $3b_1 + 9b_2 = 6b_1 + 2b_2$, so $7b_2 = 3b_1$. Substituting $b_1 = b_2 + 100$, we get

$$7b_2 = 3(b_2 + 100),$$

so $4b_2 = 300, \quad b_2 = 75, \quad$ and $\quad b_1 = 175.$

*Step 2: Verify the alternate area ratio leads to contradiction*

If $\frac{b_1 + 3b_2}{3b_1 + b_2} = \frac{3}{2}$, we get $2b_1 + 6b_2 = 9b_1 + 3b_2$, so $3b_2 = 7b_1$, which leads to a negative value for $b_2$, which is impossible. The lengths of the bases are $b_1 = 175$ and $b_2 = 75$.

*Step 3: Express the condition for equal-area division*

Let the segment of length $x$ parallel to the bases divide the trapezoid into two regions of equal area. The square of the length of such a segment is given by

$$x^2 = \frac{b_1^2 + b_2^2}{2}.$$

$$x^2 = \frac{175^2 + 75^2}{2} = \frac{30625 + 5625}{2} = \frac{36250}{2} = 18125.$$

*Step 4: Compute the final result*

We need to find the greatest integer that does not exceed $x^2/100$.

$$\frac{x^2}{100} = \frac{18125}{100} = 181.25.$$

The greatest integer that does not exceed $181.25$ is

$$\lfloor 181.25 \rfloor = 181.$$

Final Answer: The final answer is $\boxed{181}$.

# B  Template guided trajectory-level reward design

**Prompt Design of the Template (LLM-as-a-Verifier)**

You are given a long chain-of-thought (CoT) response to a challenging math problem. Your task is to summarize the response into a structured sequence of reasoning steps that can serve as a clear and guided template for use by a smaller model.

**Problem:** [problem]

**Response:** [response]

**Instructions:**
Please summarize the response as a concise list of reasoning steps, each capturing a distinct part of the thought process. These may include restating the problem, defining variables, constructing mathematical models, performing calculations, verifying results, handling different cases, correcting mistakes, and drawing the final conclusion. Focus on preserving the logical flow while keeping each step clear and concise.

Here are a few template examples you should strictly follow:

[Template Example 1]

[Template Example 2]

[Template Example 3]

Write your answer below.

# C Additional Experimental Setups

## C.1 ReasonFlux-PRM Training

For ReasonFlux-PRM training, we initialize from the off-the-shelf Qwen2.5-1.5B-Instruct and Qwen2.5-7B-Instruct models [54], serving as our 1.5B-scale and 7B-scale ReasonFlux-PRM backbones. We then train ReasonFlux-PRM on the OpenThoughts-114K [53] collection of datasets containing rich, model-generated thinking trajectories paired with their corresponding final responses.

OpenThoughts-114k is a publicly available synthetic reasoning dataset comprising 114,000 high-quality examples across four domains: mathematics, science, code, and puzzles. Each example includes a problem statement, a thinking trajectory generated by the Deepseek-R1, and a corresponding model response. The dataset was constructed by curating prompts from existing datasets, such as AI-MO/NuminaMath-CoT [22] for math, DeepMind/code-contests [25] for code, and camel-ai/chemistry [20] for science. We utilize the model-generated thinking trajectories and final responses from the datasets as raw training data. Subsequently, we assign step-level and trajectory-level rewards based on our specific reward design, as detailed in Section 4.1.

We follow our detailed description in Section 4.1 to train with the step-level reward. In addition, to train with the template-guided trajectory-level reward, we first randomly sample 1000 problem-response samples from OpenThoughts-114k, and prompt GPT-4o to extract the reasoning template from each CoT sample using the prompt in Section B. For each problem-template pair, we choose Qwen2.5-7B-Instruct as our generator $\pi_\theta$ and generate $N = 5$ responses that attempt to solve the problem while adhering to the reasoning template. The ground truth trajectory-level reward is then computed as the average accuracy across the 5 responses, as shown in Eq. 9. We then combine the step-level and trajectory-level rewards to obtain the ground truth reward values for the 1000 samples, and train ReasonFlux-PRM to learn these reward values using the joint training objective in Eq. 10. To train our reward model, we use a learning rate of 1e-5 and train for 3 epochs.

## C.2 Downstream Tasks

For offline data selection and subsequent supervised fine-tuning, we follow the exact experimental setup described in Appendix A.1 to ensure a fair comparison with baseline models. Specifically, we begin by assigning reward scores to each trajectory–response pair in OpenThoughts-114k using the designated reward model. We then rank all samples based on their aggregated reward scores and select the top 1,000 examples to serve as the training set for downstream fine-tuning.

For online policy optimization, we use a training dataset comprising 10k competition-level mathematical reasoning problems collected from MATH [13] and the DAPO [74] training set. These training data contains math problems spanning a wide range of topics, including algebra, geometry, probability, and precalculus. Our GRPO training framework is built on the original Hugging Face GRPO Trainer [57]. We train with a batch size of 32, generating 6 samples per prompt, and run training for 3 epochs. As mentioned above, the vanilla GRPO relies on a rule-based reward that evaluates only the correctness of the final answer. On the other hand, we replace the rule-based reward with the learned reward signal obtained by passing the training prompt and the policy model's output through ReasonFlux-PRM.

For the Best-of-N test-time scaling experiments, we use Qwen2.5-14B-Instruct as the generator model. Given an input problem $x$, the generator produces $N$ candidate reasoning trajectories using nucleus sampling with temperature $T = 0.3$, where $N \in \{2, 4, 8, 16\}$. Each candidate trajectory is then scored by ReasonFlux-PRM, which provides a scalar reward reflecting the trajectory's quality in terms of correctness, coherence, and reasoning structure. The final output is selected as the trajectory with the highest ReasonFlux-PRM assigned reward. We evaluate performance by measuring final-answer accuracy over the selected outputs.

# D  Additional Analyses

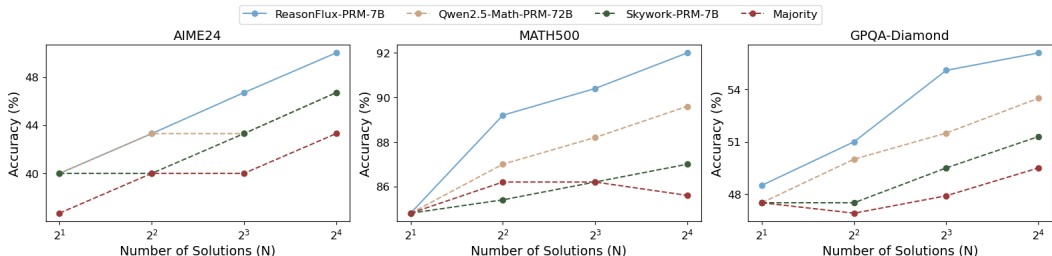

Figure 7: Test-time performance of Best-of-N selection using ReasonFlux-PRM-7B, Qwen2.5-Math-PRM-72B, and Skywork-PRM-7B across reasoning tasks. We also report results using the majority voting method.

## D.1  Best-of-N of ReasonFlux-PRM in Test-Time Scaling.

In Figure 7, we present Best-of-N selection results using ReasonFlux-PRM-7B and baseline PRMs across four reasoning tasks. For the generator model, we use the fine-tuned Qwen2.5-14B-Instruct with the same checkpoint in Section 5.1. ReasonFlux-PRM-7B consistently leads to greater accuracysuggest that larger reward models yield more accurate and informative signals for RL, thereby facilitatingor flat returns with increased sampling, ReasonFlux-PRM-7B maintains a strong upward trend, demonstrating its superior ability to identify high-quality reasoning traces.

## D.2  Scaling up ReasonFlux-PRM Model Size on Policy Optimization

Table 4: Scaling Effects of ReasonFlux-PRM model size on GRPO online policy optimization performance. Larger ReasonFlux-PRM reward models (7B vs. 1.5B) consistently yield better downstream performance on MATH500 and GPQA-Diamond across both Qwen2.5-7B-Instruct and DeepSeek-R1-Distill-Qwen-7B policy models.

| Policy Model | ReasonFlux-PRM Size | MATH500 | GPQA-Diamond |
|---|---|---|---|
| Qwen2.5-7B-Instruct | 1.5B | 73.8 | 30.8 |
| | 7B | 77.6 | 34.3 |
| DeepSeek-R1-Distill-Qwen-7B | 1.5B | 90.4 | 48.5 |
| | 7B | 93.8 | 51.5 |

To investigate the impact of reward model capacity, we vary the size of the ReasonFlux-PRM model used to provide rewards for GRPO-based policy optimization. As shown in Table 4, using a larger ReasonFlux-PRM model consistently improves performance across both policy models, Qwen2.5-7B-Instruct and DeepSeek-R1-Distill-Qwen-7B. Specifically, scaling ReasonFlux-PRM from 1.5B to 7B leads to a 3.8% gain on MATH500 and 3.5% on GPQA-Diamond for Qwen2.5-7B-Instruct. Likewise, for DeepSeek-R1-Distill-Qwen-7B, we observe a 3.4% improvement on MATH500 and 3.0% on GPQA-Diamond. These results indicate that larger reward models provide more accurate and informative signals for RL, thereby enabling stronger policy optimization.

## D.3  End-to-End Training with ReasonFlux-PRM (SFT+RL)

As supervised fine-tuning followed by reinforcement learning (SFT+RL) has become a dominant paradigm for aligning large language models with reasoning-intensive tasks, we are motivated to evaluate if ReasonFlux-PRM can serve as a general-purpose reward model to be effectively applied across both stages of training. Table 5 presents a comparative analysis on the Qwen2.5-7B-Instruct policy model, where we evaluate two checkpoints: (i) the original backbone model, and (ii) the same model after SFT on 1k ReasonFlux-PRM-7B selected data over the 59K raw data in s1 [40]. Both checkpoints are then further optimized with different reward signal sources during RL. The results

Table 5: Effect of ReasonFlux-PRM-7B selected supervised fine-tuning on downstream RL. We compare the original backbone model (Checkpoint 1) and the model fine-tuned on 1k ReasonFlux-PRM-7B selected data (Checkpoint 2), each evaluated under different reward signal sources.

| Policy Model (Qwen2.5-7B-Instruct) | Reward Signal Source | MATH500 |
|---|---|---|
| Checkpoint 1: Original backbone model | Rule-based | 74.0 |
| | Qwen2.5-Math-PRM-7B | 75.4 |
| | ReasonFlux-PRM-7B | 77.0 |
| Checkpoint 2: SFT on 1k ReasonFlux-PRM-7B selected data | Rule-based | 84.8 |
| | Qwen2.5-Math-PRM-7B | 87.6 |
| | ReasonFlux-PRM-7B | **89.8** |

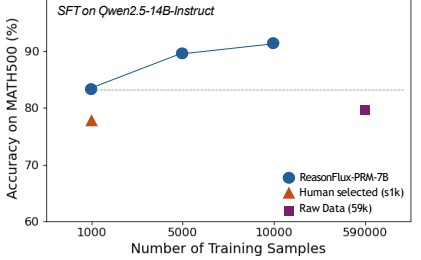 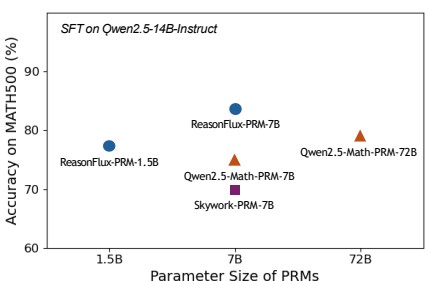

Figure 8: Effeciency Analyses on ReasonFlux-PRM-7B. **Left:** Accuracy on MATH500 improves steadily as the number of ReasonFlux-PRM-7B selected training samples increases, outperforming both human-selected (1k) and full raw data (59k) baselines with fewer total training instances. **Right:** ReasonFlux-PRM-7B achieves higher accuracy than other PRMs under 7B scale and even larger 72B scale parameter size.

demonstrate that ReasonFlux-PRM-7B consistently improves downstream performance at SFT and RL stages. We also observe that across all reward signal sources, fine-tuning on 1k ReasonFlux-PRM-7B selected data consistently improves performance over the original backbone model. Notably, the combination of ReasonFlux-PRM-7B based supervised fine-tuning and ReasonFlux-PRM-7B guided reinforcement learning yields the highest MATH500 accuracy of 89.8%, with a significant 12.8% accuracy improvement compared to the original backbone model (77.0%). These results highlight the end-to-end effectiveness of ReasonFlux-PRM as a general reward model for enhancing reasoning capabilities throughout the full training pipeline.

### D.4 Efficiency Analyses

In this section, we evaluate the efficiency of ReasonFlux-PRM-7B in both offline data selection for SFT and online RL settings by comparing the training performance and overhead under different data and reward supervision strategies.

As shown in Figure 8, the data selected by ReasonFlux-PRM-7B reduces the amount of training data required while achieving superior model performance. When fine-tuning Qwen2.5-14B-Instruct on only 1k samples selected by ReasonFlux-PRM-7B, the model outperforms the baseline trained on 59k raw trajectories by a substantial margin on MATH500. This highlights ReasonFlux-PRM 's ability to identify high-quality, informative samples that yield greater performance per data point. The result aligns with recent findings on the power of curated supervision in data-efficient post-training, and further shows that ReasonFlux-PRM-7B can outperform even human-selected samples under similar data scales.

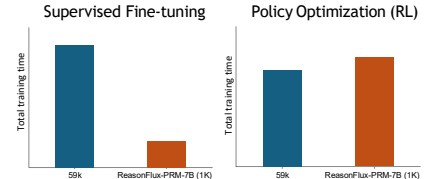

Figure 9: Time overhead of ReasonFlux-PRM- during SFT and RL stages. For SFT, we compare the training time using 1k selected samples versus the full 59k raw data. For RL training, we evaluate the overall time with/without incorporating ReasonFlux-PRM-7B.

We further investigate the overhead of incorporating ReasonFlux-PRM-7B into policy optimization using the GRPO framework. As shown in the right

Table 6: Ablation study on the $\alpha$ parameter.

| $\alpha$ | AIME24 | AIME25 | MATH500 |
|---|---|---|---|
| 0.1 | 26.7 | 6.7 | 81.2 |
| 0.8 | **40.0** | 33.3 | 83.6 |
| 1.0 | 33.3 | 33.3 | **84.8** |
| 1.5 | 33.3 | **40.0** | 83.2 |

Table 7: Ablation study on the $\beta$ parameter.

| $\beta$ | AIME24 | AIME25 | MATH500 |
|---|---|---|---|
| 0.1 | 10.0 | 6.7 | 73.6 |
| 0.3 | 13.3 | 13.3 | 74.4 |
| 0.5 | 13.3 | 6.7 | 75.2 |
| 0.8 | **20.0** | **16.7** | **76.8** |

panel of Figure 9, although ReasonFlux-PRM-7B introduces additional computation for step- and trajectory-level reward modeling, the increase in total training time remains moderate compared to standard GRPO. Crucially, this additional cost leads to consistent improvements in downstream reasoning performance, as we demonstrated in our main experiments. Our experiments on both online and offline settings above demonstrate that ReasonFlux-PRM not only improves model performance across both SFT and RL regimes, but does so with minimal computational overhead, achieving superior efficiency in reasoning-centric fine-tuning and optimization pipelines.

### D.5 Ablation Study

**Ablation on $\alpha$.** As described in Eq. 11, the parameter $\alpha$ controls the balance between step-level rewards and the trajectory-level reward during ReasonFlux-PRM's reward aggregation. To assess the impact of this weighting, we conduct an ablation study by varying $\alpha \in \{0.1, 0.8, 1.0, 1.5\}$, and use ReasonFlux-PRM-7B to select offline fine-tuning data accordingly. The Qwen2.5-14B-Instruct model is then fine-tuned on the top 1,000 selected examples and evaluated across AIME24, AIME25, and MATH500. As shown in Table 6, performance improves when more weight is placed on the trajectory-level reward. Notably, $\alpha = 1.0$ achieves the best result on MATH500, while $\alpha = 1.5$ yields the highest accuracy on AIME25. These results suggest that combining both local (step-level) and global (trajectory-level) reward signals is essential, and that moderate emphasis on trajectory-level reasoning is particularly beneficial for complex tasks. We also observe that the optimal value of $\alpha$ may be influenced by the underlying data distribution. As part of future work, we plan to make $\alpha$ learnable by introducing a lightweight neural module that dynamically adapts the weight between step-level and trajectory-level rewards based on the characteristics of each input sample.

**Ablation on $\beta$.** In Eq. 12, we introduce $\beta$ as a weighting coefficient to balance the original rule-based GRPO reward and the process-level reward provided by ReasonFlux-PRM-7B. To understand its influence, we conduct an ablation study by varying $\beta \in \{0.1, 0.3, 0.5, 0.8\}$ and applying GRPO with ReasonFlux-PRM-7B reward integration on the Qwen2.5-7B-Instruct policy model. As shown in Table 7, we evaluate the resulting models across AIME24, AIME25, and MATH500. The performance consistently improves with increasing $\beta$, indicating the effectiveness of ReasonFlux-PRM 's process-level supervision. The highest gains are achieved at $\beta = 0.8$, which yields 20.0% accuracy on AIME24, 16.7% on AIME25, and 76.8% on MATH500. The result demonstrates that a stronger emphasis on ReasonFlux-PRM rewards leads to more effective RL training.

## E   Case Study on ReasonFlux-PRM

In Case Studies 1 and 2 below, we present two responses to the same mathematical problem, one incorrect and one correct. For each response, we show the step-level and trajectory-level rewards assigned by ReasonFlux-PRM-7B. In the incorrect response, ReasonFlux-PRM-7B effectively identifies the point at which the reasoning begins to deviate by assigning significantly lower rewards to the subsequent steps. In contrast, the correct response receives consistently high rewards throughout the trajectory. Our study on the example suggests that ReasonFlux-PRM is not only capable of distinguishing between correct and incorrect solutions but also of pinpointing the precise step where an error in reasoning occurs.

# F  Additional Related Works

**Offline Data Selection for CoT Reasoning at Scale.** The quality of data has proven pivotal in the model training process [39, 77, 83, 11]. Recent studies further demonstrate that small subsets of high-quality data can outperform much larger unfiltered datasets in enhancing model reasoning abilities [41] during post-training stages such as supervised fine-tuning [30, 71, 40, 72, 1, 45, 19]. In contrast to online batch data selection methods [17, 58], which select samples based on updated model signals such as gradient norms or maximum sample loss during training, offline data selection approaches aim to select data *once* prior to the model training process. Motivated by the need for efficiency at scale, recent works have increasingly explored offline data selection as a means of curating high-quality datasets for LLMs training. Beyond simple rejection sampling, these approaches either train an additional model for data selection [64, 63], or adaptively select data based on natural language quality indicators [3], dataset diversity [61], or model-specific quality labels [31, 88]. More recently, model distillation [21, 66] has been widely adopted to leverage longer reasoning traces distilled from large-scale reasoning models as training data for improving the capabilities of downstream smaller models. Methods such as s1 [40], LIMO [72] and ReasonFlux [68] adapt smaller subsets of human-selected high-quality distilled data, enabling smaller models to perform better on sophisticated reasoning tasks compared to training on much larger quantities of raw distilled data. Building on these insights, instead of incurring additional computational costs by focusing solely on training data selection, our work extends the applicability of process reward models from traditional reward supervision to offline data selection, particularly in the context of raw model-distilled chain-of-thought reasoning trajectories [76, 9, 81]. Leveraging the step-by-step supervision capability of PRMs, we utilize them as a metric to select high-quality reasoning traces from raw "silver" distilled data [73], with the goal of improving downstream post-training performance for smaller models.

# G  Limitations and Broader Impacts

**Limitations** While ReasonFlux-PRM demonstrates robust improvements in reasoning quality across diverse mathematical and scientific benchmarks, one limitation lies in its reliance on high-quality trajectory–response pairs for training. Although ReasonFlux-PRM is designed to be general-purpose and adaptable, its current instantiation is tailored toward domains with well-structured logic (e.g., math and science), where reward signals based on alignment, quality, and coherence can be meaningfully computed. Applying ReasonFlux-PRM in more open-ended tasks such as commonsense dialogue or code generation may require redefining trajectory-level reward criteria and further tuning of the reward decomposition strategy. However, these domains fall outside the scope of this work and do not detract from the validity or strength of our conclusions in the reasoning-focused setting.

**Broader Impact** This paper introduces ReasonFlux-PRM, a trajectory-aware reward modeling framework that advances LLM reasoning by providing fine-grained supervision over both intermediate thinking steps and high-level solution strategies. This improves LLM alignment, interpretability, and efficiency in structured reasoning tasks such as math and science, while supporting both supervised fine-tuning and reinforcement learning. While our work has broad implications, we do not foresee significant negative impacts or societal concerns that warrant specific emphasis.

## Case Study 1

**Question:** If $-6 \leq a \leq -2$ and $3 \leq b \leq 5$, what is the greatest possible value of $\left(a + \frac{1}{b}\right)\left(\frac{1}{b} - a\right)$?

---

**Thinking Trajectories:**

The problem asks for the greatest possible value of the expression $\left(a + \frac{1}{b}\right)\left(\frac{1}{b} - a\right)$ given the ranges for $a$ and $b$.
[Step-level reward: 0.997]

First, let's expand the expression:
$\left(a + \frac{1}{b}\right)\left(\frac{1}{b} - a\right) = a\left(\frac{1}{b} - a\right) + \frac{1}{b}\left(\frac{1}{b} - a\right) = a \cdot \frac{1}{b} - a^2 + \frac{1}{b} \cdot \frac{1}{b} - \frac{1}{b} \cdot a = \frac{a}{b} - a^2 + \frac{1}{b^2} - \frac{a}{b}$
$= -a^2 + \frac{1}{b^2}$
[Step-level reward: 0.983]

Now we need to find the maximum value of $-a^2 + \frac{1}{b^2}$ given the constraints $-6 \leq a \leq -2$ and $3 \leq b \leq 5$.
[Step-level reward: 0.996]

Consider the terms $-a^2$ and $\frac{1}{b^2}$ separately. Since $-6 \leq a \leq -2$, we have $0 \leq (-a)^2 \leq (-(-6))^2 = 36$. Therefore, $-36 \leq -a^2 \leq -4$.
[Step-level reward: 0.503]

Since $3 \leq b \leq 5$, we have $9 \leq b^2 \leq 25$. Therefore, $\frac{1}{25} \leq \frac{1}{b^2} \leq \frac{1}{9}$.
[Step-level reward: 0.895]

We want to maximize $-a^2 + \frac{1}{b^2}$. This is equivalent to minimizing $-a^2$ and maximizing $\frac{1}{b^2}$.
[Step-level reward: 0.297]
*(This is the place where the thinking trajectory went wrong, because in order to maximize $-a^2 + \frac{1}{b^2}$, $-a^2$ needs to be maximized, not minimized. ReasonFlux-PRM assigns low reward starting from this segment within the thinking trajectory, which indicates that ReasonFlux-PRM is able to identify errors in reasoning traces.)*

The minimum value of $-a^2$ is $-36$.
The maximum value of $\frac{1}{b^2}$ is $\frac{1}{9}$.
[Step-level reward: 0.238]
*(The step-level rewards drop significantly since the last wrong step.)*

So the greatest possible value of $-a^2 + \frac{1}{b^2}$ is $-36 + \frac{1}{9}$.
Let's calculate this value: $-36 + \frac{1}{9} = -\frac{36 \times 9}{9} + \frac{1}{9} = -\frac{324}{9} + \frac{1}{9} = -\frac{324-1}{9} = -\frac{323}{9}$
[Step-level reward: 0.280]

---

**Final Response:**
Given the ranges $-6 \leq a \leq -2$ and $3 \leq b \leq 5$. We want to find the greatest possible value of the expression $\left(a + \frac{1}{b}\right)\left(\frac{1}{b} - a\right)$.
[Step-level reward: 0.995]

First, we expand the expression: $\left(a + \frac{1}{b}\right)\left(\frac{1}{b} - a\right) = a\left(\frac{1}{b} - a\right) + \frac{1}{b}\left(\frac{1}{b} - a\right) = a \cdot \frac{1}{b} - a^2 + \frac{1}{b} \cdot \frac{1}{b} - \frac{1}{b} \cdot a$
$= \frac{a}{b} - a^2 + \frac{1}{b^2} - \frac{a}{b} = -a^2 + \frac{1}{b^2}$
[Step-level reward: 0.997]

Now we need to find the maximum value of $-a^2 + \frac{1}{b^2}$ given the constraints for $a$ and $b$. Consider the term $-a^2$. Since $-6 \leq a \leq -2$, the values of $a^2$ range from $(-6)^2 = 36$ to $(-2)^2 = 4$. Therefore, the values of $-a^2$ range from $-36$ to $-4$.
[Step-level reward: 0.988]

Consider the term $\frac{1}{b^2}$. Since $3 \leq b \leq 5$, the values of $b^2$ range from $3^2 = 9$ to $5^2 = 25$. Therefore, the values of $\frac{1}{b^2}$ range from $\frac{1}{25}$ to $\frac{1}{9}$.

To maximize $-a^2 + \frac{1}{b^2}$, we need to maximize $\frac{1}{b^2}$ and minimize $-a^2$. The maximum value of $\frac{1}{b^2}$ is $\frac{1}{9}$, which occurs when $b = 3$. The minimum value of $-a^2$ is $-36$, which occurs when $a = -6$.

[Step-level reward: 0.108]
*(The low score here aligns with the low score in the model thinking trajectories.)*

Thus, the greatest possible value of $-a^2 + \frac{1}{b^2}$ is $-36 + \frac{1}{9}$.

$$-36 + \frac{1}{9} = -\frac{36 \times 9}{9} + \frac{1}{9} = -\frac{324}{9} + \frac{1}{9} = -\frac{324 - 1}{9} = -\frac{323}{9}$$

Final Answer: The final answer is $\boxed{-\dfrac{323}{9}}$

## Case Study 2

**Question:** If $-6 \leq a \leq -2$ and $3 \leq b \leq 5$, what is the greatest possible value of $\left(a + \frac{1}{b}\right)\left(\frac{1}{b} - a\right)$?

---

**Thinking Trajectories:**

The problem asks for the greatest possible value of the expression $\left(a + \frac{1}{b}\right)\left(\frac{1}{b} - a\right)$ given the ranges

for $a$ and $b$.
[Step-level reward: 0.998]

[Trajectory-level reward: 0.197]
*(As the final answer is incorrect, ReasonFlux-PRM assigns a low trajectory-level reward to the overall trajectory-response data.)*

Let $E = \left(a + \frac{1}{b}\right)\left(\frac{1}{b} - a\right)$.
We can expand the expression:
$E = a\left(\frac{1}{b}\right) + a\left(-a\right) + \frac{1}{b}\left(\frac{1}{b}\right) + \frac{1}{b}\left(-a\right)$
$E = \frac{a}{b} - a^2 + \frac{1}{b^2} - \frac{a}{b}$
$E = -a^2 + \frac{1}{b^2}$
[Step-level reward: 0.985]

We need to find the maximum value of $E$ given the constraints $-6 \leq a \leq -2$ and $3 \leq b \leq 5$.
[Step-level reward: 0.998]

Since $E = -a^2 + \frac{1}{b^2}$, we need to minimize $a^2$ and maximize $\frac{1}{b^2}$.
[Step-level reward: 0.974]

For $a$, the range is $-6 \leq a \leq -2$. The minimum value of $a^2$ occurs when $a$ is closest to 0. The values of $a$ are negative, so the minimum value of $a^2$ occurs when $a$ is the least negative, which is $-2$.
When $a = -2$, $a^2 = (-2)^2 = 4$.
[Step-level reward: 0.962]

For $b$, the range is $3 \leq b \leq 5$. The maximum value of $\frac{1}{b^2}$ occurs when $b$ is the smallest.
When $b = 3$, $\frac{1}{b^2} = \frac{1}{3^2} = \frac{1}{9}$.
[Step-level reward: 0.985]

So, the maximum value of $E$ occurs when $a^2$ is minimized and $\frac{1}{b^2}$ is maximized.
Maximum $E = -a^2 + \frac{1}{b^2} = -4 + \frac{1}{9}$.
[Step-level reward: 0.818]

To add these values, we need a common denominator:
Maximum $E = -4 + \frac{1}{9} = -\frac{4 \times 9}{9} + \frac{1}{9} = -\frac{36}{9} + \frac{1}{9} = -\frac{36-1}{9} = -\frac{35}{9}$.
[Step-level reward: 0.815]
Let's double check the calculations.
The expression is $\left(a + \frac{1}{b}\right)\left(\frac{1}{b} - a\right)$.
Expand the expression: $a \cdot \frac{1}{b} + a \cdot (-a) + \frac{1}{b} \cdot \frac{1}{b} + \frac{1}{b} \cdot (-a)$
$= \frac{a}{b} - a^2 + \frac{1}{b^2} - \frac{a}{b}$
$= -a^2 + \frac{1}{b^2}$
[Step-level reward: 0.980]

We want to maximize $-a^2 + \frac{1}{b^2}$.
Since $a^2 \geq 0$, $-a^2 \leq 0$.
Since $b^2 > 0$, $\frac{1}{b^2} > 0$.
The maximum value of $\frac{1}{b^2}$ is $\frac{1}{3^2} = \frac{1}{9}$, which occurs when $b = 3$.
We need to find the minimum value of $a^2$ in the range $[-6, -2]$.
The minimum value of $a^2$ occurs at the boundary of the interval, which is $a = -2$.
When $a = -2$, $a^2 = (-2)^2 = 4$.
[Step-level reward: 0.927]

We need to maximize $\frac{1}{b^2}$ in the range $[3, 5]$.
The maximum value of $\frac{1}{b^2}$ occurs at the boundary of the interval, which is $b = 3$.
When $b = 3$, $\frac{1}{b^2} = \frac{1}{3^2} = \frac{1}{9}$.
[Step-level reward: 0.989]

So the maximum value of the expression is $-a^2 + \frac{1}{b^2} = -4 + \frac{1}{9} = -\frac{36}{9} + \frac{1}{9} = -\frac{35}{9}$.

---

**Final Response:**
Given the ranges for $a$ and $b$: $-6 \leq a \leq -2 \ 3 \leq b \leq 5$
[Step-level reward: 0.958]

We want to find the greatest possible value of the expression $\left(a + \frac{1}{b}\right)\left(\frac{1}{b} - a\right)$.
[Step-level reward: 0.999]

First, we expand the expression:

$$\left(a + \frac{1}{b}\right)\left(\frac{1}{b} - a\right) = a\left(\frac{1}{b}\right) + a\left(-a\right) + \frac{1}{b}\left(\frac{1}{b}\right) + \frac{1}{b}\left(-a\right)$$

$$= \frac{a}{b} - a^2 + \frac{1}{b^2} - \frac{a}{b}$$

$$= -a^2 + \frac{1}{b^2}$$

[Step-level reward: 0.998]

To find the greatest possible value of this expression, we need to minimize $a^2$ and maximize $\frac{1}{b^2}$.
[Step-level reward: 0.987]

For the term $-a^2$, we want to minimize $a^2$. Since $a^2 \geq 0$ for all real numbers $a$, the minimum value of $a^2$ occurs when $a = 0$. However, the given range for $a$ is $[-6, -2]$, so $a = 0$ is not within the range. The minimum value of $a^2$ in the given range occurs at the boundary of the interval, which is $a = -2$. When $a = -2$, $a^2 = (-2)^2 = 4$.
[Step-level reward: 0.967]

For the term $\frac{1}{b^2}$, we want to maximize $\frac{1}{b^2}$. Since $b^2 > 0$ for all real numbers $b \neq 0$, the maximum value of $\frac{1}{b^2}$ occurs when $b^2$ is the smallest. The smallest value of $b^2$ in the given range occurs when $b$ is the largest in absolute value within the interval $[3, 5]$, which is $b = 3$. When $b = 3$, $b^2 = 3^2 = 9$, and $\frac{1}{b^2} = \frac{1}{9}$.
[Step-level reward: 0.667]

Now, we substitute the minimum value of $a^2$ and the maximum value of $\frac{1}{b^2}$ into the expanded expression:

$$-a^2 + \frac{1}{b^2} = -4 + \frac{1}{9}$$

To combine these terms, we find a common denominator:

$$-4 + \frac{1}{9} = -\frac{4 \times 9}{9} + \frac{1}{9} = -\frac{36}{9} + \frac{1}{9} = -\frac{36 - 1}{9} = -\frac{35}{9}$$

[Step-level reward: 0.993]

Thus, the greatest possible value of the given expression is $-\frac{35}{9}$.
[Step-level reward: 0.988]

Final Answer: The final answer is $\boxed{-\dfrac{35}{9}}$
[Step-level reward: 0.999]

---

[Trajectory-level reward: 0.949]
*(As the final answer is correct, ReasonFlux-PRM assigns a high trajectory-level reward to the overall trajectory-response data.)*

