# OpenReview forum: "ReasonFlux-PRM: Trajectory-Aware PRMs for Long Chain-of-Thought Reasoning in LLMs"
_NeurIPS.cc/2025/Conference — NeurIPS 2025 poster_

### Official Review · Reviewer_vTjq · 2025-06-24

**Clarity:** 3
**Significance:** 3
**Originality:** 3
**Rating:** 5
**Confidence:** 4

**Summary:**

The paper proposes TAP (Trajectory-Aware Process Reward Model), a new framework for evaluating and supervising intermediate reasoning steps in LLM. Unlike existing PRMs focused on final outputs, TAP assigns fine-grained rewards to entire reasoning trajectories using step-level signals and a trajectory-level check via a proxy policy model. The proposed model could be used in both offline and online reward supervision. It achieves over 22% accuracy gains on hard reasoning benchmarks like AIME and GPQA compared to prior reward models.

**Questions:**

Same as the Weaknesses section:


1. The paper uses GPT-4o as the expert model. Could the authors provide an estimate of the associated cost, and discuss whether it could be replaced with an open-source model? Would that affect performance?
2. DeepSeek-R1 reported that PRMs did not work well in their setting, whereas the proposed method appears effective. Could the authors provide further analysis on this discrepancy?
3. I am also curious whether the performance gain from this method, compared to rule-based judges, increases or decreases as model size scales up.
4. I suggest the authors add experiments comparing their method to step-wise LLM-as-judge baselines.

**Ethical Concerns:**

["NO or VERY MINOR ethics concerns only"]

**Final Justification:**

In this paper, the authors propose TAP (Trajectory-Aware Process Reward Model), a new framework for evaluating and supervising intermediate reasoning steps in LLM. Unlike existing PRMs focused on final outputs, TAP assigns fine-grained rewards to entire reasoning trajectories using step-level signals and a trajectory-level check via a proxy policy model.

During the rebuttal, the authors have shown their method's cost is cheap and also could use the open-source model to replace. Furthermore, they conduct some experiments on the 14B and 32B model, which shows the effectiveness of the proposed TAP. Based on these strengths, I decide to raise my score.

**Limitations:**

yes

**Paper Formatting Concerns:**

NAN

**Quality:**

3

**Strengths And Weaknesses:**

## Strengths

1. The paper is well-structured and the motivation is clear.
2. It advances PRM research from post-hoc answer checking to process-aware supervision; it demonstrates that well-scored trajectories can outperform human curation and enhance reinforcement learning.
3. The proposed method achieves strong performance in both offline and online reward supervision settings.

## Weaknesses

1. The paper uses GPT-4o as the expert model. Could the authors provide an estimate of the associated cost, and discuss whether it could be replaced with an open-source model? Would that affect performance?
2. DeepSeek-R1 reported that PRMs did not work well in their setting, whereas the proposed method appears effective. Could the authors provide further analysis on this discrepancy?
3. I am also curious whether the performance gain from this method, compared to rule-based judges, increases or decreases as model size scales up.
4. I suggest the authors add experiments comparing their method to step-wise LLM-as-judge baselines.

---

> ### Author Rebuttal · Authors · 2025-07-31
>
> We sincerely thank the reviewer for your constructive suggestions! We address each of your concerns below.
>
> ---
> ## **[Q.1] Cost analysis of GPT-4o & Open-source Alternatives.**
>
> Thank you for the insightful question regarding expert model cost. **Our trajectory-level reward method is model-agnostic and can be seamlessly applied with any expert model, including open-source alternatives.** As a reply:
>
> 1. We first provide the associated cost of using GPT-4o as the expert model.
> 2. We then demonstrate the effectiveness of our method when adopting two additional open-source models as the expert.
>
> **1. GPT-4o Cost.**
> ||Token Count|Price (per 1 mil tokens)|Cost|
> |-|-|-|-|
> |Input|3,589,427|$2.5|$8.97|
> |Output|343,864|$10|$3.44|
> |Total|3,933,291|-|**$12.41**|
>
> The table above reports the primary cost of using GPT-4o as the expert model. Our total cost on the expert model is **less than $15**. We highlight that our "template-guided" trajectory reward method is **cost-efficient** for the following reasons:
> - In our TAP model training, we utilize only a relatively small dataset (**1,000 samples**) to prompt GPT-4o for template generation, which is sufficient to demonstrate effectiveness in downstream reward modeling.
> - The expert model GPT-4o is used solely to **generate concise, structured templates** (typically under 500 tokens), significantly reducing token costs compared to directly leveraging the expert model to produce full-length CoT reasoning traces and solutions.
>
> **2. Effectiveness of Open-source Alternatives.**
> Regarding performance impact, as noted in Lines 190–197, the policy model uses templates generated by the expert model to produce $N$ responses, which are then used to compute trajectory rewards.
>
> Consequently, the choice of expert model will influence TAP training quality: **a more powerful and reliable expert model provides more robust trajectory reward signals during PRM training, leading to stronger performance.**
>
> To better address the reviewer's question, **we newly adopt DeepSeek-R1 and QwQ-32B as the common open-source expert models for the 1,000 template generation**. We follow the same training setups in our paper. We report the new comparison results on offline data selection and SFT:
> |Expert Model|MATH500|GPQA-Diamond|
> |-|-|-|
> |GPT-4o| 84.8|47.5|
> |QwQ-32B|84.2| 46.0|
> |DeepSeek-R1|86.4| 49.5|
>
> - From the table, we observe that TAP remains effective when using open-source expert models. Notably, as DeepSeek-R1 is more powerful on reasoning tasks, adopting it as the expert model surpasses the original GPT-4o with higher quality trajectory rewards.
> - Thus, **TAP’s effectiveness is not tied to a single proprietary expert model, and it can generalize well to other high-quality open-source alternatives**.
>
> ---
> ## **[Q.2] Justification on DeepSeek-R1 Findings.**
> We thank the reviewer for asking this insightful question. We justify from three key perspectives:
>
> **1. More advanced PRM Design:**\
> In DeepSeek-R1 [1], the authors noted that several early-stage PRMs (e.g., [2,3]) are ineffective for complex reasoning tasks. However, our PRM is designed and trained fundamentally different from the following:
> - **Effectiveness on Long CoT Data.** The adopted PRMs in [1] were primarily trained and designed for simpler tasks, where the PRM training data typically contained short outputs with minimal step-level information. In contrast, **DeepSeek-R1** is trained to generate much longer and more complex CoT outputs. As a result, these earlier PRMs often failed when required to provide step-level reward supervision for such long CoT outputs.
>
>     In contrast, **TAP is specifically designed and trained on advanced reasoning tasks with much longer and more complex CoT data (OpenThoughts-114K)**, which includes not only models’ final responses but also rich intermediate thinking trajectories.
>
> - **Overlook of Intermediate Thinking Trajectories.** In our preliminary studies in `Section 3`, we found that models’ intermediate thinking trajectories are integral for effective process supervision. However, existing PRMs largely overlook this rich intermediate information and fail to provide reliable supervision on these trajectories.
>
>     In contrast, **TAP is the first trajectory-aware PRM to explicitly incorporate models’ intermediate thinking trajectories**, enabling it to provide robust and fine-grained process rewards throughout the reasoning process.
>
> - **Process Reward Design.** The earlier PRMs in [1] often failed to provide fine-grained step-level rewards, which led to issues such as reward hacking.
>
>     In contrast, **TAP** provides both step-level and trajectory-level reward supervision, capturing both fine-grained and holistic reasoning quality in long CoT data. This dual-level design offers a more precise and reliable optimization objective for complex reasoning tasks.
>
>
> **2. Broader applicability on TAP:**\
> We also want to highlight that the core idea and contribution of TAP is to **support more general and comprehensive reward modeling scenarios**. Beyond its application in **RL policy optimization** as mentioned in [1], TAP also provides process reward supervision for tasks such as **offline data selection** and **inference-time scaling**.
>
>
>
> **3. Evolution of the process reward-modeling:**
> Lastly, we want to briefly mention that several recent works [4,5,6] have shown that richer process-reward signals lead to improved performance. This suggests that more advanced PRMs can potentially overcome the limitations of earlier, outdated PRMs used in prior studies.
>
> **References:**\
> [1] DeepSeek-R1: Incentivizing Reasoning Capability in LLMs via Reinforcement Learning. \
> [2] Solving math word problems with process- and outcome-based feedback.\
> [3] Math-shepherd: Verify and reinforce LLMs step-by-step without human annotations. \
> [4] Rewarding Progress: Scaling Automated Process Verifiers for LLM Reasoning. \
> [5] Inference-Time Scaling for Generalist Reward Modeling. \
> [6] Process Reinforcement through Implicit Rewards.
>
> ---
> ## **[Q.3] Scaling Trend of Performance Gain vs. Model Size.**
>
> Thank you for the questions regarding the scalability of our method. We have conducted **additional experiments on larger policy models** right after our original submission.
>
> Below, we show that TAP is scalable to larger policy models with improved reward modeling:
>
> - **Experiment Setups:** We follow the same experimental setup as in `Section 5.2` to run GRPO. As an extension, we integrate TAP into larger policy models, including **Qwen-2.5-14B-Instruct** and **Qwen-2.5-32B-Instruct**.
> - **New Results:**
>     |Policy Model|Reward Signal Source|AIME24|AIME25|MATH500|
>     |-|-|-|-|-|
>     |**Qwen2.5-14B-Instruct**| Rule-based |36.7|26.7|84.6|
>     ||**TAP (ours)**|**43.3**|**40.0**|**86.8**|
>     |**Qwen2.5-32B-Instruct**| Rule-based |53.3|46.7|92.4|
>     ||**TAP (ours)**|**56.7**|**53.3**|**95.2**|
>
>
>     From the table above, TAP outperforms the rule-based judges baseline across both 14B and 32B policy models.
>
>     Our new experiment demonstrates that **TAP scales effectively to different sizes (7B, 14B, and 32B) policy models** and **delivers consistent performance gains**.
>
> - We thank the reviewer for their question and will ensure to incorporate our new experiment results into our paper for a better empirical demonstration of our method.
>
> ---
> ## **[Q.4] Additional Comparison to Step-wise LLM-as-judge.**
> We thank the reviewer for the helpful suggestion. In response:
> - **Existing Step‑wise Baselines:** We first want to highlight that several PRM baselines compared in our paper, such as Qwen2.5-Math-PRM-72B [1], already employ step‑wise LLM‑as‑judge signals during training.
> - **TAP's Novelty:** Unlike PRM baselines directly utilizing LLM-as-judge to obtain reward signals, **TAP leverages an expert LLM as the verifier to enforce trajectory‑level coherence via template alignment.**
> - **Additional Experiments:** We conduct additional experiments to **compare TAP with a direct step-wise LLM-as-judge baseline** on two test time scaling tasks.
>     - **Experimental Setup:** Following prior works [2,3], we use GPT-4o as the judge LLM to score and rerank $N = 16$ candidate responses using pure step-wise LLM-as-judge aggregation (i.e., averaging GPT-4o judged quality scores across all steps). Additional setups on best-of-N TTS are the same as `Section 5.2` in our paper.
>     - **Results:**
>         |Method|GPQA-Diamond@16|MATH500@16|
>         |-|-|-|
>         |LLM-as-Judge|51.5|90.4|
>         |**TAP**|**56.6**|**92.2**|
>
>         From the results, TAP consistently outperforms pure step‑wise LLM-as-Judge baseline on both GPQA-Diamond and MATH500.
>
> Due to the rebuttal time constraints, we present a simplified baseline here. We will incorporate these results into our manuscript and evaluate more advanced LLM‑as‑judge variants in future work.
>
>
> **References:**\
> [1] The Lessons of Developing Process Reward Models in Mathematical Reasoning. \
> [2] Let's Verify Step by Step. \
> [3] Judging LLM-as-a-Judge with MT-Bench and Chatbot Arena.

---

> > ### Comment · Reviewer_vTjq · 2025-08-01
> >
> > Thanks for the authors' detailed reply, which has resolved my concerns. Hence, I decide to raise my score, and hope other reviewers could also see the effectiveness and real-world value of this work.

---

> ### Author Response · Authors · 2025-08-01
> **Thanks for raising score!**
>
> Dear Reviewer,
>
> Thank you for your positive feedback and for recognizing the real-world value of our work!
>
> We are pleased that our rebuttal has addressed your concerns. We will incorporate the results and analyses from the rebuttal to further strengthen our paper.
>
> Warm Regards,
>
> The Authors

---

### Official Review · Reviewer_N1UW · 2025-06-30

**Clarity:** 3
**Significance:** 3
**Originality:** 3
**Rating:** 4
**Confidence:** 2

**Summary:**

This paper introduces TAP, a reward modeling framework designed to assess the quality of the reasoning trajectories. The authors argue that conventional PRMs, only trained on final outputs, are not suitable for evaluating the trajectories. TAP incorporates both step-level rewards and trajectory-level rewards to tackle this challenge. TAP is shown to outperform existing PRMs.

**Questions:**

* Why does scoring based on final responses perform similarly to PRMs trained on final outputs? Doesn’t this go against the main claim that trajectory-level supervision is essential?
* Appendix D.1 You say TAP has low overhead—can you show reward computation time alone? Is it just that other parts dominate total cost?
* Appendix D.2 You show that trajectory-level reward helps, but is step-level reward really needed? Especially r_coh—why assume adjacent steps must be similar? Can you show ablations using only step-level or only trajectory-level rewards?

**Ethical Concerns:**

["NO or VERY MINOR ethics concerns only"]

**Final Justification:**

Thanks to the authors for their detailed explanations. While I still think that the method is a bit excessive (e.g., minimal influence of $r^{coh}$), my other concerns are resolved. Specifically, I was unsure how to explain Table 1, due to the modest gap between the PRM and the human-curated data. The authors' clarification in [Q.1] has provided a clear understanding of the table. Consequently, I have decided to maintain my current score.

**Limitations:**

yes

**Paper Formatting Concerns:**

One minor formatting issue: On page 2, the page number appears to overlap slightly with the main text. It would be good to adjust the positioning to avoid visual clutter in the camera-ready version.

**Quality:**

3

**Strengths And Weaknesses:**

Strength
* The paper tackles a timely and significant gap in PRM research, aligned with the trajectory-based outputs of modern frontier LLMs.
* TAP consistently outperforms prior PRMs and even human-curated datasets.

Weakness
* Despite claims in Appendix D.1, TAP’s design—especially the LLM-based scoring, template generation, and trajectory decoding—seems over-engineered for many practical applications.
* Insufficient ablation analysis. While the authors analyze α (the trajectory reward weight), there’s no clear ablation showing the effect of removing each component of the reward (e.g., removing r_coh or r_align entirely).
* Although Table 1 is used to argue that conventional PRMs are miscalibrated on reasoning trajectories, the support is not entirely convincing—performance gaps are modest, and the diagnosis lacks granularity.
* TAP can only be applied post-hoc after a full trajectory is generated, limiting its use for online reranking or early stopping.

---

> ### Author Rebuttal · Authors · 2025-07-31
>
> Thank you for your thorough feedback! Below, we address each of the concerns.
>
> ---
> ## **[W.1] TAP's Method Design.**
> > "TAP’s design—especially the LLM-based scoring, template generation, and trajectory decoding—seems over-engineered for many practical applications."
>
> Thank you for the insightful question. While we acknowledge that TAP incorporates multiple design components and may appear "over-engineered" at first glance, we would like to justify our motivation and design choices for the following reasons:
>
> - In `Section 3`, we thoroughly evaluate existing PRMs and identify several key shortcomings from our preliminary experiments. To address these issues, we designed TAP with targeted components, each aimed at mitigating specific limitations of prior PRMs.
> - As an illustration, we integrate (i) **LLM-based scoring** to provide Fine-grained logical correctness at the step level; (ii) **Template generation** to ensure trajectory-level reward supervision; and (iii) **Trajectory decoding** for practical integration into RL and test-time inference.
>
> In our following replies to the reviewer's questions, we will conduct new ablation studies to verify the effectiveness of each component in TAP’s design.
>
> ---
> ## **[W.2 & Q.3] Ablation Analyses on TAP.**
> > "Is step-level reward really needed? Can you show ablations using only step-level or only trajectory-level rewards?"
>
> Thank you for asking about the effectiveness of each component in TAP. To better illustrate our design choices, we present several ablation studies below.
>
> **1. Ablations on TAP's step-level rewards choices: alignment, quality, and coherence ($r^{align}, r^{qual}, r^{coh}$).**
>
> **Setups:** We follow the same training setups in `Section 5` and `Appendix C.1` for the ablation studies. For downstream evaluation, we choose offline data selection and report the SFT performance.
> |Method|MATH500|GPQA-Diamond|
> |-|-|-|
> |w/o $r^{align}$|82.4|43.4|
> |w/o $r^{qual}$|81.8|41.9|
> |w/o $r^{coh}$|84.4|46.0|
> |**TAP (ours)**|**84.8**|**47.5**|
>
> The results show that:
> - Removing the quality and coherence scores during TAP training leads to noticeably suboptimal performance.
> - while the alignment score has relatively less impact compared to the other two-step reward supervision.
> - Motivated by these ablations, we plan to explore a lightweight version of TAP in the future that includes only selected step-level rewards tailored to specific application needs.
>
>
>
> **2. Ablations on TAP's step-level and trajectory-level rewards ($r^{step}, r^{final}$).**
>
> We follow the same experimental setup as above to train different TAP variants, each excluding either step-level or trajectory-level rewards.
>
> **Note:** We increase the training data for trajectory-level rewards from 1k to 10k for training stability.
> |Method|MATH500|GPQA-Diamond|
> |-|-|-|
> |w/o $r^{final}$|82.8|43.9|
> |w/o $r^{step}$|83.4|44.9|
> |**TAP (ours)**|**84.8**|**47.5**|
>
> Our ablation study here shows that removing either the trajectory-level ($r^{final}$) or step-level ($r^{step}$) reward degrades performance, with trajectory-level rewards having a greater impact. This underscores the necessity of jointly training TAP with both reward signals.
>
>
> We thank the reviewer for the suggestion and will incorporate our new ablations into the revision to better demonstrate the contribution of each component in TAP.
>
> ---
> ## **[W.3] Analysis on the Shortcomings of Conventional PRMs.**
> >"Although Table 1 is used to argue that conventional PRMs are miscalibrated on reasoning trajectories, the support is not entirely convincing—performance gaps are modest, and the diagnosis lacks granularity."
>
> We thank the reviewer for raising this question. In response:
>
> **1. More Detailed Interpretation on Table 1.**
> - As part of our preliminary experiments, Table 1 serves as one component of our diagnosis of existing PRMs’ shortcomings on the trajectory-response data.
> - Although larger PRMs, such as Qwen2.5-Math-PRM-72B, show a relatively modest gap compared to the human-curated baseline, smaller PRMs like Skywork-PRM-7B and Math-Shepherd-PRM-7B exhibit significant gaps and even fail to outperform random selection.
>
> **2. Additional Analyses and Diagnosis on Existing PRMs.** \
> We kindly point out that our argument on conventional PRMs is miscalibrated on reasoning trajectories, which has also been supported by multiple other complementary analyses:
> - **Distributional Evidence:** In `Section 3`, we show that existing PRMs (e.g., Qwen2.5-Math-PRM-72B) exhibit significant overlap in score distributions between trajectories distilled from stronger (DeepSeek-R1) vs. weaker (Gemini) oracles.
> - **Granular Diagnostics in Appendix:** In `Appendix A.2`, we supplement Table 1 with additional analyses highlighting the **formatting mismatch between thinking trajectories and final responses.**.
>
> We thank the reviewer for the question and will update our paper with a more detailed diagnostic analysis to improve both clarity and logical flow.
>
> ---
> ## **[W.4] Justifications on Online Reranking and Early Stopping.**
> > "TAP ... limiting its use for online reranking or early stopping."
>
> We appreciate the reviewer's observation. While TAP is indeed applied post-hoc on the trajectory-response data in our current design, we would like to clarify that:
>
> **1. Consistent with common PRM's objective.**\
> In `Section 2 (Preliminaries)`, we explicitly define the problem formulation of process reward modeling on trajectory-response data. **This objective is consistent with the formulations adopted by all other existing PRMs cited in our paper**.
>
> **2. Primary Objective of TAP on General Applications.**\
> As the main focus of our paper, TAP is designed as a **general-purpose PRM** to support high-quality data selection for offline training, reward modeling for online policy optimization in RL, and test-time scaling.
>
> **3. Extension on Online Reranking and Early Stopping.**\
> We kindly interpret the reviewer’s intention as suggesting that TAP should provide robust process rewards during **token-level generation**. While current PRMs like TAP are primarily designed for **sequence-level process supervision**, we agree that extending TAP to support token-level supervision, including online reranking and early stopping, **is a promising research direction**. We will incorporate such studies in future work.
>
> ---
> ## **[Q.1] Trajectory-level Supervision.**
>
> We thank the reviewer for the insightful question. In Table 1, we observed that existing PRMs trained only on final responses perform better when scoring final responses rather than full trajectories.
>
> We argue that this observation highlights the importance of trajectory-level supervision for the following reasons:
>
> >"Doesn’t this go against the main claim that trajectory-level supervision is essential?"
>
> **1. Importance of Trajectory-Response CoT Data.**
> As current advanced large reasoning models mostly adopt the output format of thinking trajectories followed by final responses, and these full reasoning chains are often used as training data for downstream distillation, providing supervision only on the final response is insufficient and incomplete.
>
> >"Why does scoring based on final responses perform similarly to PRMs trained on final outputs?"
>
> **2. Suboptimal Performance without Trajectory-level Supervision.**
> In Table 1, existing PRMs perform better when used to select only final responses compared to selecting entire trajectory-response data. However, regardless of selection based on entire trajectory-response or response-only, there remains a performance gap between PRM-selected data and the human-curated baseline.
>
> In addition, as analyzed in `Appendix A.2`, thinking trajectories differ significantly from final responses, which makes it challenging for existing PRMs to provide reliable rewards on these intermediate trajectory steps.
>
> **3. Empirical Validation of TAP with the Trajectory-level Reward.**
> To address the problem mentioned above, we design TAP with the template-guided trajectory-level reward to access that **whether the overall problem-solving strategy encoded in the model’s thinking trajectory reliably leads to correct solutions in final responses.**
>
> As later shown in Table 2, TAP outperforms not only existing PRMs applied to model final responses (row: "on model responses") but also the strongest human-curated baseline.
>
> ---
> ## **[Q.2] Computation Overhead of TAP.**
> > "Appendix D.1 You say TAP has low overhead—can you show reward computation time alone? Is it just that other parts dominate the total cost?"
>
> We thank the reviewer for asking about the computation overhead of our method. In response:
>
> **1. Time Overhead Report on Reward Computation**
>
> We provide a time overhead report, including both the reward computation and the total time overhead.
>
> **Setup:** We run TAP-enhanced GRPO under the GPU settings in our paper and report the average time per step over 500 training steps. The results are shown below:
> |Base Model|Method|Avg. Reward Computation Time/Step (s)| Avg. Total time/Step (s)| Percentage|
> |-|-|-|-|-|
> |Qwen2.5-7B-Instruct|TAP|7.42|183.75| 4.0%|
> |DeepSeek-R1-Distill-Qwen-7B| TAP|11.45|484.71|2.4%|
>
> The table shows that  TAP’s reward computation takes **< 5%** of the overall training time per step. This demonstrates that **incorporating TAP into RL policy optimiation adds marginal overhead to the overall runtime**.
>
> **2. Detailed Explanation of TAP’s High Efficiency**
> In `Section D.1`, we attribute TAP’s efficiency to two key factors:
>
> - **Data Efficiency:** TAP reduces training data size by offline selecting high-quality examples (e.g., 1k TAP-selected data outperforms both raw 59k data and human-curated 1k data). This significantly lowers both human annotation costs and overall SFT training time.
> - **Model Efficiency:** TAP uses far fewer parameters while achieving superior reward modeling performance compared to 7B and even 72B PRMs, thereby reducing deployment and inference costs.

---

> > ### Comment · Reviewer_N1UW · 2025-08-06
> >
> > Thanks to the authors for their detailed explanations. While I still think that the method is a bit excessive (e.g., minimal influence of $r^{coh}$), my other concerns are resolved. Specifically, I was unsure how to explain Table 1, due to the modest gap between the PRM and the human-curated data. The authors' clarification in [Q.1] has provided a clear understanding of the table.
> > Consequently, I have decided to **maintain my current score.**

---

> ### Author Response · Authors · 2025-08-06
>
> Dear Reviewer N1UW,
>
> Thank you for maintaining your positive rating of our submission! We are pleased that our clarifications have addressed your other concerns (e.g., explanation of Table 1).
>
> We also appreciate your insightful question on TAP's design complexity. In our revised manuscript:
> - **In-Depth Analyses of TAP:** We will incorporate the ablation studies from the rebuttal, along with a detailed discussion, to provide a clearer rationale for each component of TAP—particularly the role and functionality of $r^{\text{coh}}$.
>
> - **Lightweight Design of TAP:** As suggested by the reviewer, we will explore simplified versions of TAP to improve usability and accessibility while retaining effectiveness.
>
> Thank you again for your constructive feedback, which helps strengthen our paper. We will incorporate analyses and results from our rebuttal into our revision.
>
> Warm regards,
>
> The Authors

---

### Official Review · Reviewer_wGca · 2025-07-02

**Clarity:** 3
**Significance:** 3
**Originality:** 4
**Rating:** 5
**Confidence:** 4

**Summary:**

This paper proposes a new process reward model (PRM) for improving reasoning, especially on hard math problems. They identify an important limitation of existing process reward models: that they focus on scoring only the final response and not on the intermediate reasoning trajectories. They set up an experiment where they show that existing process reward models cannot distinguish reliably between the intermediate reasoning trajectories from different models, even when the PRM can distinguish between final responses.
They then define TAP which is a trajectory aware process reward model. TAP combines a number of separate step level rewards and a whole trajectory reward together. The step level rewards include: alignment scores between trajectory steps and final answers, LLM-as-a-judge of step quality, and a coherence between steps score. The whole trajectory reward takes the trajectory and summarises it with a strong LLM and then provides this as a prompt to an LLM, and evaluates if the LLM gets the question correct with this summary. They show that TAP outperforms baseline PRMs for offline data selection of fine-tuning data, online reinforcement learning with GRPO for smaller reasoning models, and for test time n-best list reranking.

**Questions:**

In strengths and weaknesses

**Ethical Concerns:**

["NO or VERY MINOR ethics concerns only"]

**Final Justification:**

A good paper with interesting results

**Limitations:**

Limitations in Appendix which was in supplementary materials and I did not read it.

**Quality:**

4

**Strengths And Weaknesses:**

Strengths:
	1) The paper is clear and well written.
	2) The problem is important: improving reasoning is currently one of the most interesting and impactful areas of AI research
	3) They have identified an important drawback of current PRMs. Although the final answer is clearly important, the trajectories are valuable and should be part of the reward. Good trajectories will lead to better answers even if they contain self corrections, redundant reasoning and branching logic.
	4) The have suggested a significantly novel approach to rewarding both trajectories and answers in one reward model.
	5) The have demonstrated that TAP outperforms baselines across 3 very different uses of a reward model: data selection, reinforcement learning, and n-best list reranking.

Weaknesses:
	1) I am not completely convinced that TAP leads to significant improvements over the baseline reward models. I am slightly concerned that for reinforcement learning they used one of the smaller models (7B). RL is the most important application of a PRM and it would have been nice to show improvements over the 14B or the 72B baseline which were used in other experiments in this paper. The data selection experiment again was good, but why stop at 1000 examples? That being said - the experiments cover 3 good test sets and 3 diverse tasks and show improvements on all three so I do think TAP works - the question remaining for me is - how well?
	2) (not major) I don't think that the score distribution graphs tell us as much as is claimed. The score distributions between Gemini and Deepseek are the same for trajectories and different for final answers yes, but the authors should at least discuss if these  distribution differences are meaningful. Are the Gemini answers really so much worse? This is not made clear or justified
	3) (minor) It is odd that they report mostly the same table of results in Table 1 and then again in Table 2. Perhaps they should combine into one table and reorder discussion somewhat to reflect this.

---

> ### Author Rebuttal · Authors · 2025-07-31
>
> Thank you for your thorough comments and suggestions! Below, we address each of your concerns.
>
> ---
> ## **[W.1] Scalability on Larger Policy Models &  Data Selection Examples.**
> We thank the reviewer for the thoughtful questions regarding our experiments. In response:
> 1. We first present **new RL experiments** incorporating TAP on **14B and 32B policy models**.
> 2. We then **justify our intention of using 1,000 examples** in the data selection experiments.
> 3. We further **expand the training data size to 3k and 5k** to demonstrate the effectiveness and scalability of our method.
>
> **1. New RL Experiments with Larger Policy Models.**\
> We have conducted additional experiments on larger policy models right after the original submission. We present the updated experiments below.
>
> **Setups:** We follow the same experimental setups as `Section 5.2` to run GRPO. For new add-ups:
> - Larger Policy Model: We integrate TAP into larger policy models, including **Qwen-2.5-14B-Instruct**, **Qwen-2.5-32B-Instruct**.
> - Larger PRM Baseline: We compare TAP with a larger PRM baseline, including the **Qwen2.5-Math-PRM-72B.**
>
> **Results:**
> |Policy Model|Reward Signal Source|AIME24|AIME25|MATH500|
> |-|-|-|-|-|
> |**Qwen2.5-14B-Instruct**| Rule-based |36.7|26.7|84.6|
> ||Qwen2.5-Math-PRM-72B|36.7|33.3|85.2|
> ||**TAP (ours)**|**43.3**|**40.0**|**86.8**|
> |**Qwen2.5-32B-Instruct**| Rule-based |53.3|46.7|92.4|
> ||Qwen2.5-Math-PRM-72B|50.0|46.7|92.2|
> ||**TAP (ours)**|**56.7**|**53.3**|**95.2**|
>
> - From the table above, TAP outperforms baselines, including rule-based methods and Qwen2.5-Math-PRM-72B, **across both 14B and 32B policy models**.
> - This demonstrates that **TAP scales effectively to different sizes (7B, 14B, and 32B) policy models** and delivers greater performance gains compared to even larger existing PRMs.
>
> Our results above show a consistent improvement trend of TAP across scales. Due to rebuttal time and resource limits, we leave additional experiments on larger models (e.g., 72B-scale policy models) to future work.
>
> **2. Rationale for 1,000 Examples in Data Selection.**\
> The **primary reason** for using 1,000 examples in offline data selection is to **ensure a fair comparison with the strongest baseline (i.e., the 1k high-quality human-curated data in s1k[1])**.
> - s1k [1] shows that a small set of 1,000 human-expert-selected CoT examples can outperform much larger amounts of raw data.
> - Given the high cost of human annotation, we explore **a new perspective** of applying PRMs for automated high-quality data selection. **Under the same 1,000-example experiment condition**, we evaluate whether TAP can efficiently outperform human-curated data in improving downstream SFT performance for smaller models (as shown in Table 2).
>
>
> **3. Scaling Training Data to 3k and 5k for Better Validation.**\
> To better address the reviewer’s question, we extend the data selection experiment in `Section 5.1` by applying TAP to select **3k and 5k samples** (from the same pool of 59k raw data traces) and fine-tune the generator (Qwen2.5-14B-Instruct) under the identical setup.
>
> - The results are reported below:
>     |SFT Data Size (TAP)|AIME24|MATH500|GPQA-Diamond|
>     |-|-|-|-|
>     |1k|40.0|84.8|47.5|
>     |3k|40.0|85.4|49.0|
>     |5k|43.3|86.2|49.5|
> - The results above show that **increasing the size of TAP-selected data generally improves downstream performance**, with a clear upward trend across all three benchmarks.
>
>
> We thank the reviewer for these insightful questions and will incorporate the suggestions and new experimental results into our paper.
>
> ---
> ## **[W.2 (not major)] Deeper Analysis on Score Distribution.**
>
> We appreciate the reviewer's observation and agree that clarifying the meaning of the score distributions is important. Below,
>
> 1. We **first** present a data quality analysis comparing DeepSeek-R1 and Gemini Flash Thinking, demonstrating that DeepSeek-R1 indeed generates higher-quality thinking trajectories and thus merits higher process rewards.
> 2. We **then** discuss why our observation, as shown in **Figure 2** of the paper, is meaningful and how it motivated us to design our PRM.
>
>
> **1. Analyses on the Quality of Generated Thinking Trajectories.**\
> We first follow [1] and utilize **1,000 thinking trajectories CoT data distilled separately from DeepSeek-R1 and Gemini Flash Thinking API**. We then perform supervised fine-tuning on a downstream model (**Qwen-2.5-14B**) using each of the two 1k datasets, while keeping all other settings identical. Finally, we evaluate the resulting models on **four downstream tasks** (same as **Table 1** of our paper) and report the results below:
> | SFT Data Source | AIME 24 | AIME 25 | MATH 500 | GPQA-Diamond|
> |-|-|-|-|-|
> |1k instances from **Gemini Flash Thinking**|33.3|33.3|78.8|41.4|
> |1k instances from **DeepSeek-R1**|43.3|40.0|80.2|44.9|
>
> - From the results, we observe a substantial performance gap between the two training data sources. Fine-tuning with distilled CoT data from **DeepSeek-R1** significantly improves downstream accuracy, **indicating that DeepSeek-R1 indeed produces higher-quality thinking trajectories**.
> - We further conducted a closer examination and case studies of Gemini’s “Flash Thinking” traces and found that they are noisier and more verbose than traces from DeepSeek-R1, often containing exploratory but ultimately irrelevant and incorrect reasoning steps.
> - Our observation is also consistent with findings from prior works on reasoning traces, such as [2, 3].
>
>
> **2. Discussion on Why Distribution Differences are Meaningful.**
> - From **Figure 2 (left)** of our paper, we observe that on thinking trajectories, existing PRMs' output score distributions between DeepSeek-R1 and Gemini Flash Thinking are **largely overlapped**.
> - Given that DeepSeek-R1 generally produces higher-quality thinking traces, as demonstrated in our analyses above, this highlights a **key limitation of current PRMs**: their inability to reliably distinguish and reward high-quality thinking trajectories.
> - Motivated by the observations above, we argue that a comprehensive PRM should not only assess final answers but also **provide robust rewards on thinking trajectories, enabling the selection of complete high-quality CoT traces for downstream applications**. As demonstrated in **Figure 4**, our TAP effectively differentiates the quality of thinking trajectories from different sources (i.e., via the score distribution differences), which in turn leads to improved downstream SFT performance.
>
>
> We thank the reviewer for the insightful question and will incorporate the above explanations into our paper to improve clarity.
>
> ---
> ## **[W.3 (minor)] Tables Results Formatting.**
> We thank the reviewer for the helpful suggestion! Our intention in reporting results in both **Table 1** and **Table 2** was to serve distinct purposes:
>
> - **Table 1 (Problem Diagnosis):** Reports the sub-optimal performance of existing PRMs to highlight their limitations and motivate our work.
> - **Table 2 (Solution Efficacy):** Compares our proposed method against other PRMs and human-curated baselines to demonstrate its effectiveness.
>
> We will incorporate the reviewer’s suggestion to reorder our discussion for improved clarity and readability of the paper.
>
> ## **References:**
> [1] s1: Simple test-time scaling.\
> [2] LIMO: Less is More for Reasoning.\
> [3] DeepSeek-R1: Incentivizing Reasoning Capability in LLMs via Reinforcement Learning.

---

> > ### Comment · Area_Chair_UsAx · 2025-08-05
> > **Please post your response**
> >
> > Dear Reviewer wGca,
> >
> > This is a gentle reminder to post your response. The deadline for the author-reviewer discussion period is approaching. The authors have responded to your reviews and also to others' reviews. Please have an open discussion with the authors about your reviews and whether your concerns have been addressed.
> >
> > Best,
> >
> > AC

---

> ### Comment · Reviewer_wGca · 2025-08-05
> **response to rebuttal**
>
> Dear authors,
>
> I commend you for responding in full to my concerns with significantly scaled up experiments. This strengthens the paper considerably in my opinion. My recommendation is already accept and but I can increase my confidence in the overall score.

---

> > ### Author Response · Authors · 2025-08-05
> >
> > Dear Reviewer,
> >
> > Thank you for your positive feedback. We will incorporate the results and analyses from the rebuttal to strengthen our paper.
> >
> > Warm Regards,
> >
> > The Authors

---

### Official Review · Reviewer_Wmgb · 2025-07-05

**Clarity:** 3
**Significance:** 2
**Originality:** 2
**Rating:** 4
**Confidence:** 4

**Summary:**

This paper identifies the challenge of supervising long, unstructured "thinking trajectories" from advanced LLMs, a gap left by existing PRMs focused on shorter, structured reasoning. The paper propose a new Process Reward Model, TAP, which combines alignment, quality, and coherence scores to provide fine-grained rewards for these long trajectories. The goal is to use this reward model to improve the complex reasoning abilities of frontier models through reinforcement learning.

**Questions:**

The related work section would benefit from discussing more recent and relevant Reasoning-PRM evaluation paradigms [1,2].
[1] R-PRM: Reasoning-Driven Process Reward Modeling
[2] Inference-Time Scaling for Generalist Reward Modeling

Hyperparameter Analysis:  Given the monotonic improvement shown in Table 5, have you try an experiment for β=1 (pure reward from PRM)

line318 should be Appendix F

**Ethical Concerns:**

["NO or VERY MINOR ethics concerns only"]

**Final Justification:**

Author's detailed response address some of my concern, i have changed the overall score to 4

**Limitations:**

yes

**Quality:**

2

**Strengths And Weaknesses:**

# Strengths
The paper's main strength is its excellent problem formulation. It correctly identifies a critical and underexplored area: supervising the raw, lengthy "thinking trajectories" of frontier models, moving beyond the limitations of PRMs.

# Weaknesses
* Methodological Complexity vs. Ablation: The method's complexity, with its multiple reward components and encoders, is not justified by a proper ablation study. It is unclear if all parts are necessary, making the design feel over-engineered.
* Counter-intuitive Training Data Size: The claim that the PRM is trained on only 1,000 samples (Appendix C.1) is counter-intuitive and lacks justification.
* Questionable Experiment Setup:
  + It is not explained how the Qwen2.5-PRM baseline, with its 4096-token context limit, evaluated long trajectories.
  + Strange RL Results: The reinforcement learning results in Table 3 appear flawed. The performance of baselines, especially the rule-based reward, is significantly underestimated, eg. refer to [1].
  + Suboptimal Best-of-N Evaluation: The Best-of-N evaluation uses a restrictive sampling temperature (T=0.3), limiting diversity (commonly using 0.6~1). Majority Voting seems to be underestimated. Also recomend to include the upper-bound baseline like Pass@N.

---

> ### Author Rebuttal · Authors · 2025-07-31
>
> We appreciate the reviewers' efforts and detailed feedback. We address each of your concerns below.
>
> ---
> ## **[W.1] Methodological Complexity vs. Ablation.**
> Thank you for the insightful question. For better elaboration of our TAP design:
>
> **1. Motivation on TAP Method Design.**
> - In `Section 3`, we first thoroughly evaluate existing PRMs and identify several key shortcomings from our preliminary experiments. To address these issues, we designed TAP with targeted components, each aimed at mitigating specific limitations of prior PRMs.
> - As an illustration, we integrate (i) **LLM-based scoring** to provide fine-grained logical correctness at the step level; (ii) **Template generation** to ensure trajectory-level reward supervision; and (iii) **Trajectory decoding** for practical integration into RL and test-time inference.
>
> **2. Ablations on TAP Method Components.**
>
> **[Ablation Study 1]** TAP's step-level rewards choices: alignment, quality, and coherence ($r^{align}, r^{qual}, r^{coh}$).
>
> **Setups:** We follow training setups in `Section 5` and `Appendix C.1` for the ablation studies. For downstream evaluation, we choose offline data selection and report the SFT performance.
> |Method|MATH500|GPQA-Diamond|
> |-|-|-|
> |w/o $r^{align}$|82.4|43.4|
> |w/o $r^{qual}$|81.8|41.9|
> |w/o $r^{coh}$|84.4|46.0|
> |**TAP (ours)**|**84.8**|**47.5**|
>
> The results show that:
> - Removing the quality and coherence scores during TAP training leads to noticeably suboptimal performance.
> - While the alignment score has relatively less impact compared to the other two-step reward supervision.
> - Motivated by these ablations, we plan to explore a lightweight version of TAP in the future that includes only selected step-level rewards tailored to specific application needs.
>
> **[Ablation Study 2]** TAP's step-level and trajectory-level rewards ($r^{step}, r^{final}$).
>
> We follow the same experimental setup as above to train different TAP variants, excluding either step-level or trajectory-level rewards. We increase the training data for trajectory-level rewards from 1k to 10k for better training.
> |Method|MATH500|GPQA-Diamond|
> |-|-|-|
> |w/o $r^{final}$|82.8|43.9|
> |w/o $r^{step}$|83.4|44.9|
> |**TAP (ours)**|**84.8**|**47.5**|
>
> Our ablation study shows that removing either the trajectory-level ($r^{final}$) or step-level ($r^{step}$) reward degrades performance, with trajectory-level rewards having a greater impact. This underscores the necessity of jointly training TAP with both reward signals.
>
> We will incorporate our new ablations into the revision to better demonstrate the contribution of each component in TAP.
>
> ---
> ## **[W.2] Training Data Size.**
>
> We apologize for the confusion regarding TAP's training data size. As we stated in `Appendix C.1`:
>
> > Line 407: We train TAP on the OpenThoughts-114K collection of datasets.
>
> > Line 419: In addition to training with the template-guided trajectory-level reward, we first randomly sample 1000 problem-response samples from OpenThoughts-114k.
>
> TAP is trained on the **full 114K data**, while **the 1,000 samples are used for trajectory-level reward training**, since:
> - Our experiments show that 1k samples suffice for robust trajectory-level training.
> - Trajectory-level reward signals are more computationally expensive to obtain than step-level rewards.
>
> We will refine our description of TAP's training settings to improve readability and clarity.
>
> ---
> ## **[W.3.1] Explanation on Qwen2.5-PRM Baseline.**
>
> Thank you for the question. The baseline PRMs, Qwen2.5-Math-PRM-7B and Qwen2.5-Math-PRM-72B, indeed support a maximum context length of 4096.
>
> For the evaluated trajectory-response data **over 4096 tokens**:
> - We first separate the data into thinking trajectories and final responses, respectively.
> - For each part, the PRM computed step-level rewards by segmenting steps (using "\n\n" separators).
> - Then we leverage the PRMs to assign step rewards for them separately and then average the entire step rewards as the final reward.
> - These step-level scores were then aggregated (via mean) together.
> - Note that if either the trajectory or response individually exceeded 4096 tokens, we truncated the excess tokens to fit within the PRM’s context window, which we found to be a necessary preprocessing step for compatibility with Qwen2.5-Math-PRM.
>
> We also **highlight** that, unlike existing PRMs limited to scoring final responses, TAP extends supervision to long CoT data by incorporating both intermediate thinking trajectories and final responses, better aligning with frontier reasoning models.
>
> We thank the reviewer and will incorporate the detailed settings above into our paper to improve clarity.
>
> ---
> ## **[W.3.2] RL Results Justification.**
>
> Thank you for the careful review and the helpful reference [1]. In response to the TAP's performance on RL policy optimization:
>
> **1. Updated RL Performance on TAP:**
> In our early-stage experiments, due to submission time constraints, we settled the context length of the policy model (DeepSeek-R1-Distill-Qwen-7B) to a reduced value of 2800. We recognize that this could potentially hinder the generation.
>
> To address this issue, we **reset the context length to 16,384** and conducted experiments in `Section 5.2`. We also included an extra downstream task, GPQA-Diamond. The new updated results are reported below:
> |Policy Model|Reward Signal Source|AIME24|AIME25|MATH500|GPQA-Diamond|
> |-|-|-|-|-|-|
> |**Qwen2.5-7B-Instruct**|Rule-based|12.9|11.1|73.6|32.7|
> ||Qwen2.5-Math-PRM-7B|12.9|13.3|74.8|32.4|
> ||**TAP**|**16.3**|**17.1**|**77.2**|**34.9**|
> |**DeepSeek-R1-Distill-Qwen-7B**|Rule-based|50.2|38.3|89.6|47.1|
> ||Qwen2.5-Math-PRM-7B|51.2|40.8|92.8|49.1|
> ||**TAP**|**54.6**|**44.2**|**94.8**|**51.6**|
>
> **2. TAP on Larger Policy Model:**
> We also conducted additional experiments of TAP on larger policy models right after our submission. We present the updated experiments for the reviewer's reference:
>
> **Setups:** We follow the same experimental setups as `Section 5.2` to run GRPO.
> - **Larger Policy Model**: Qwen-2.5-14B-Instruct and Qwen-2.5-32B-Instruct.
> - **Larger PRM Baseline**:  Qwen2.5-Math-PRM-72B.
>
> |Policy Model|Reward Signal Source|AIME24|AIME25|MATH500|
> |-|-|-|-|-|
> |**Qwen2.5-14B-Instruct**| Rule-based |36.7|26.7|84.6|
> ||Qwen2.5-Math-PRM-72B|36.7|33.3|85.2|
> ||**TAP**|**43.3**|**40.0**|**86.8**|
> |**Qwen2.5-32B-Instruct**| Rule-based |53.3|46.7|92.4|
> ||Qwen2.5-Math-PRM-72B|50.0|46.7|92.2|
> ||**TAP**|**56.7**|**53.3**|**95.2**|
>
> - **Consistent Conclusion with Original Submission:** Our experiments across 7B, 14B, and 32B policy models affirm our original claim in `Section 5.2` regarding the effectiveness of TAP on RL.
>
> We will incorporate the new experiments above and the related work [1] mentioned by the reviewer into our paper revision to improve clarity.
>
> ---
> ## **[W.3.3] Best-of-N Evaluation.**
>
> We thank the reviewer for the helpful suggestion. In response:
> - We conduct new experiments below using a proper temperature setting of $T=0.7$.
> - We add the upper-bound pass@N and the majority@N metrics with $N = 8$.
> |Method|AIME24|MATH500|GPQA-Diamond|
> |-|-|-|-|
> |TAP Accuracy@8 (T=0.3)|46.7|91.2|55.6|
> |TAP Accuracy@8 (T=0.7)|46.7|91.4|56.6|
> |Majority@8 (T=0.7)|40.0|87.8|51.0|
> |TAP pass@8 (T=0.7)|60.0|93.6|59.1|
>
> From the results on the fine-tuned Qwen2.5-14B-Instruct policy model:
> - Increasing $T$ from 0.3 to 0.7 yields improved performance for the policy model.
> - Consistent with the results in `Section 5.2`, TAP shows better improvements over the updated majority voting baseline and the new pass@8 metric.
>
> We hope our new experiments provide a more comprehensive best-of-N evaluation of TAP and will incorporate results into our paper.
>
> ---
> ## **[Q.1] Additional Related Works.**
> We thank the reviewer for the suggestion and agree that incorporating these recent works on reasoning-focused PRMs strengthens our related work section.
>
> As a **brief discussion** on the relationship between TAP and each work:
> - **Relation to R-PRM [1]:** R-PRM improves step-level reward modeling via reasoning-driven analysis and preference optimization. TAP enables **offline data selection from a trajectory-aware perspective** and focuses more specifically on the **trajectory–response form of long CoT data**.
> - **Relation to DeepSeek-GRM [2]:** DeepSeek‑GRM uses a point‑wise GRM paradigm for inference‑time scaling. While TAP also supports best‑of‑N selection, we employ a different training and reward design, providing more dense process‑level feedback rather than only scalar final‐answer scores.
>
> We thank the reviewer for the suggestions and will revise our related work section to **include a detailed discussion of these two works and cite them appropriately** in our revision.
>
> ---
> ## **[Q.2] Hyperparameter Analysis on $\beta=1$.**
>
> Thank you for the question. We indeed conduct experiments **with $\beta=1$**, and the updated experiment results is shown below:
>
> |β|AIME24|AIME25|MATH500|
> |-|-|-|-|
> |0.1|10.0|6.7|73.6|
> |0.3|13.3|13.3|74.4|
> |0.5|13.3|6.7|75.2|
> |0.8|20.0|16.7|76.8|
> |1.0|20.0|16.7|76.6|
>
> - As the reviewer noted, increasing $\beta$ from 0.1 to 0.8 yields a clear monotonic gain, demonstrating that **stronger emphasis on TAP’s process reward steadily boosts policy performance**.
> - At $\beta=1.0$, we observe several tasks plateau. We attribute this to **the need for outcome‑level anchoring**:  By relying exclusively on the learned PRM reward at  $\beta=1.0$, we drop the direct correctness signal that **anchors learning to the final-task objective.** Losing such direct correctness feedback might potentially introduce noise or bias.
> - Our ablations indicate that **$\beta=0.8$ is the “sweet spot”**, offering the best trade‑off between TAP's rich trajectory‑level supervision and enough outcome reward to stabilize learning.
>
> ## **References:**
> [1] R-PRM: Reasoning-Driven Process Reward Modeling.\
> [2] Inference-Time Scaling for Generalist Reward Modeling.

---

> > ### Comment · Reviewer_Wmgb · 2025-08-04
> >
> > Thank you for your response, i have changed my rating

---

> ### Author Response · Authors · 2025-08-05
>
> Dear Reviewer,
>
> Thank you for your positive feedback. Your feedback has been invaluable in improving our paper.
>
> Warm Regards,
>
> The Authors

---

### Decision · Program_Chairs · 2025-09-17

**Decision:**

Accept (poster)

**Comment:**

The paper introduces TAP (Trajectory-Aware PRM), a novel reward modeling framework designed to evaluate multi-step reasoning trajectories of large language models. Unlike existing PRMs focusing on final responses, TAP incorporates both step-level (alignment, quality, coherence) and trajectory-level supervision to provide fine-grained feedback. The authors demonstrate TAP’s effectiveness in three applications: offline data selection for distillation, online reinforcement learning, and test-time reranking via best-of-N strategies. Experiments on AIME and GPQA benchmarks show a 22.7% performance improvement over baseline PRMs, validating its utility for supervising complex reasoning in frontier models like OpenAI-o1 and DeepSeek-R1.

Reviewers acknowledge the paper’s timely and impactful contribution to PRM research. The strong problem formulation addresses an underexplored gap in supervising lengthy, unstructured reasoning trajectories from frontier LLMs. The framework’s versatility across offline/online settings and its consistent outperformance of baselines, including human-curated data, are highlighted as key strengths. The significance of advancing PRM research from final-output focus to process-aware supervision is also emphasized.

On the other hand, reviewers point out several weaknesses. Critiques include insufficient ablation studies to justify the complexity of TAP’s components, and limited scalability analysis. The RL experiments’ focus on smaller models and the restrictive sampling temperature in best-of-N evaluation raise questions about broader applicability (Reviewer wGca, Wmgb). Some reviewers request cost analysis for GPT-4o dependency and comparisons to step-wise LLM-as-judge baselines.

In general, the paper tackles a timely and important problem with a novel framework that shows promising results across multiple applications. While it has limitations needing revision, its strengths in problem relevance, methodological innovation, and empirical performance outweigh the weaknesses. Thus, the paper is recommended for acceptance with minor revisions.

During the rebuttal, the authors demonstrated that their method is cost-effective and can be replaced with an open-source model. Furthermore, they conduct some experiments on larger policy models, and ablations on TAP method components, which shows the effectiveness of the proposed TAP. The effective rebuttal leads to the increase of average rating.